# Towards Reliable LLM-based Robot Planning via Combined Uncertainty Estimation

**Shiyuan Yin**[1], **Chenjia Bai**[2][*], **Zihao Zhang**[1], **Junwei Jin**[1], **Xinxin Zhang**[1],
**Chi Zhang**[2], **Xuelong Li**[2]

[1] School of Artificial Intelligence, Henan University of Technology
[2] Institute of Artificial Intelligence (TeleAI), China Telecom

## Abstract

Large language models (LLMs) demonstrate advanced reasoning abilities, enabling robots to understand natural language instructions and generate high-level plans with appropriate grounding. However, LLM hallucinations present a significant challenge, often leading to overconfident yet potentially misaligned or unsafe plans. While researchers have explored uncertainty estimation to improve the reliability of LLM-based planning, existing studies have not sufficiently differentiated between epistemic and intrinsic uncertainty, limiting the effectiveness of uncertainty estimation. In this paper, we present **C**ombined **U**ncertainty estimation for **R**eliable **E**mbodied planning (CURE), which decomposes the uncertainty into epistemic and intrinsic uncertainty, each estimated separately. Furthermore, epistemic uncertainty is subdivided into task clarity and task familiarity for more accurate evaluation. The overall uncertainty assessments are obtained using random network distillation and multi-layer perceptron regression heads driven by LLM features. We validated our approach in two distinct experimental settings: kitchen manipulation and tabletop rearrangement experiments. The results show that, compared to existing methods, our approach yields uncertainty estimates that are more closely aligned with the actual execution outcomes. The code is at https://github.com/Firesuiry/CURE.

## 1 Introduction

Large Language Models (LLMs) have shown exceptional versatility [1, 2, 3], as evidenced by their ability to handle a wide array of tasks. These tasks include answering complex questions [4], solving mathematical problems [5], generating computer code [6], and performing sophisticated reasoning during inference [7]. When robots try to execute tasks instructed by language descriptions, they can leverage LLM capabilities to interpret task instructions in natural language [8, 9], apply the common sense reasoning of LLMs to understand their environment [10], and formulate high-level action plans based on the robot's abilities and available resources.

However, a significant challenge with current LLMs is their propensity to hallucinate [11]—i.e., to confidently generate outputs that appear plausible but are actually incorrect and practically infeasible. [12]. This misplaced confidence in erroneous outputs presents a considerable obstacle to LLM-based planning in robotics. Additionally, natural language instructions in real-world environments often contain inherent or unintentional ambiguity from human instructors [13]. And following a flawed plan with excessive confidence may lead to undesirable or even unsafe actions. Consequently, offering an uncertainty estimation for the planning model prior to execution is desirable to reflect the level of confidence in their plans, allowing for halting execution or seeking human assistance in cases of high uncertainty.

---

[*]Correspondence to Chenjia Bai <baicj@chinatelecom.cn>

39th Conference on Neural Information Processing Systems (NeurIPS 2025).

Previous studies on uncertainty in reinforcement learning (RL) typically distinguish two components of model uncertainty: epistemic uncertainty and intrinsic uncertainty [14]. However, in research on robotics LLM planners, these uncertainty components have not been analyzed at a fine-grained level. This lack of comprehensive analysis can lead to inaccurate assessments of model capabilities, such as misattributing uncertainty caused by environmental randomness to the model itself, thereby underestimating its true abilities [15]. Furthermore, when environmental changes occur, it is challenging to isolate and update intrinsic uncertainty, hindering rapid adaptation to new conditions.

The empirical evidence from cognitive research reveals that clear task descriptions significantly enhance decision accuracy by providing a precise framework for cognitive processing and reducing the cognitive load associated with ambiguity [16]. Conversely, vague objectives tend to introduce substantial cognitive dissonance, thereby increasing hesitation and the likelihood of errors [17]. Similarly, the familiarity of tasks plays a crucial role in execution efficiency. Familiar tasks, which have been ingrained through repeated exposure and practice, allow for streamlined cognitive and motor processes, thereby optimizing performance [18]. In contrast, unfamiliar tasks necessitate additional cognitive resources for schema construction and adaptation, thus introducing greater uncertainty and potential inefficiencies. These cognitive insights indicate that epistemic uncertainty can be decomposed into components related to task description clarity and task familiarity, enabling more precise uncertainty modeling for LLM task planning.

In this paper, we separately estimate planning uncertainty and introduce a new method for uncertainty estimation. First, we examine epistemic uncertainty in planning tasks, further breaking it down into two factors: task clarity and task familiarity. Second, we model intrinsic uncertainty as the expected success rate of a given plan. A lower expected success rate indicates a higher level of intrinsic uncertainty as it implies that despite following the plan, there still exist factors within the environment that may lead to failure. To evaluate these uncertainties, we use multi-layer perceptrons regression heads driven by LLM features to estimate task clarity and expected success rate, while a Random Network Distillation (RND) network is employed to assess task familiarity. In addition, we propose a slower yet more precise task clarity evaluation method based on LLM-query. By combining these components, we provide a comprehensive uncertainty assessment for LLM-based planning, enhancing the accuracy of uncertainty estimation.

We compare the proposed algorithm with state-of-the-art uncertainty estimation techniques in robot planning, as well as widely-used uncertainty estimation methods in LLMs. The experiments include mobile manipulator in kitchen tasks and tabletop rearrangement tasks. The results demonstrate that the uncertainty estimates produced by the proposed method were the most accurate and comprehensive, showing the strongest correlation with the actual execution result.

**Our main contributions are summarized as follows: First**, we propose a novel uncertainty evaluation method CURE that separately assesses epistemic uncertainty and intrinsic uncertainty to better align model confidence with task success. **Second**, we present an effective method for assessing task similarity in task planning, leveraging RND to enhance the foundation for uncertainty estimation. **Third**, our experiments with a mobile manipulator in kitchen and tabletop rearrangement scenarios demonstrate that the proposed algorithm significantly enhances the accuracy of uncertain estimation, outperforming existing methods in embodied planning and LLM uncertain estimation.

## 2 Preliminaries

In this section, we introduce the core concepts underlying our proposed CURE framework, including the definitions of uncertainty, the problem formulation and an overview of RND.

**Definitions of uncertainty.** Epistemic uncertainty is uncertainty due to a lack of knowledge or information. It is reducible in principle, as it can be decreased by gathering more data or building better models. It stems from ignorance. Intrinsic uncertainty (also called aleatoric uncertainty) is the inherent randomness or variation in a system itself. It is irreducible, meaning it cannot be eliminated even with perfect information or infinite data. It stems from the fundamental chaosof the world.

**Problem statement.** We consider a manipulation problem in which a high-level, free-form human instruction $I$ is given (e.g., Give me something to drink). The observation $O$ includes an instruction pertaining to the current environment. An LLM planner decomposes the task into an object-centric executable task plan $A$. The central problem investigated in this work is how to generate an uncertainty

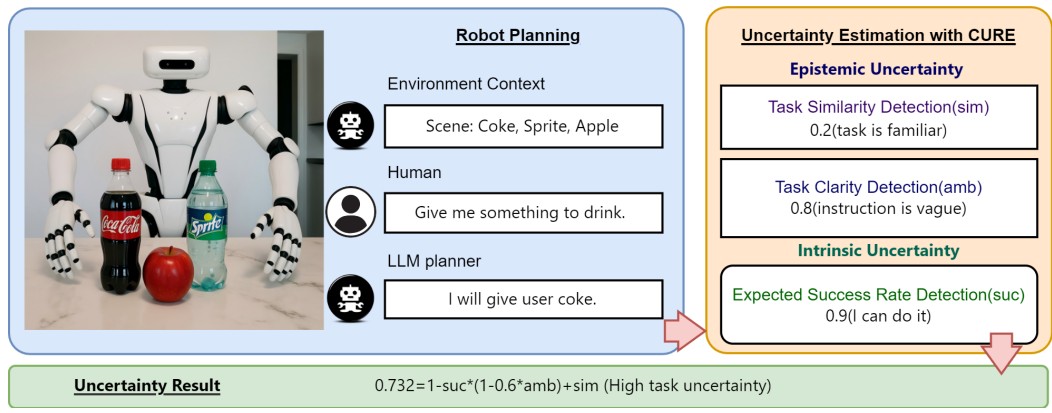

Figure 1: **Overview of the proposed uncertainty-aware LLM planning framework.** Given a natural language instruction (e.g., Give me sth to drink) and environmental context (e.g., Coke, Sprite, Apple), the LLM planner generates a high-level plan (e.g., I will give user Coke). Our framework estimates the overall planning uncertainty using CURE module, which decomposes uncertainty into epistemic and intrinsic components. Epistemic uncertainty encompasses task similarity and task clarity, while intrinsic uncertainty is represented by the expected success rate of the generated plan. The final uncertainty score then guides the decision to proceed, halt, or request clarification, thereby enhancing planning reliability in uncertain or ambiguous scenarios.

estimate $U$ that quantifies the planner's confidence in the task planning process. For the convenience in subsequent processing, we input $I$, $O$, and $A$ into the LLama3.2-8B model to obtain the hidden activations of the last token from the last layer, which serve as the task encoding vector $T$.

Let $S = \{s_1, s_2, \ldots, s_n\}$ represent the set of observed success rates, and $U = \{u_1, u_2, \ldots, u_n\}$ denote the corresponding set of planning uncertainty estimates, where $n$ is the number of samples.

The objective is to maximize the statistical dependence between $S$ and $U$, as measured by Spearman's rank correlation coefficient $\rho$. Formally, we aim to solve:

$$\max \rho(S, U) \tag{1}$$

where $\rho(S, U)$ is defined as:

$$\rho(S, U) = 1 - \frac{6 \sum_{i=1}^{n} d_i^2}{n(n^2 - 1)}, \quad d_i = \text{rank}(s_i) - \text{rank}(u_i). \tag{2}$$

Here, $\text{rank}(s_i)$ and $\text{rank}(u_i)$ denote the ranks of $s_i$ and $u_i$ within their respective datasets.

**Random Network Distillation.** Random Network Distillation (RND) is an approach initially utilized to measure the novelty of a given state, aiming to encourage exploration for rarely-visited states in large state space in RL. The core idea behind RND is to use a randomly initialized neural network, referred to as the *target network*, which is kept fixed during training. A separate *predictor network* is then trained to predict the output of the target network, typically via a regression loss. The target network's weights are not updated throughout the training process, and its outputs remain fixed. This fixed target network is essentially a source of "random" or "unknown" knowledge, and the predictor network attempts to minimize the difference between its predictions and the outputs of the target network. The error in the predictor network's predictions provides a measure of how well the predictor is able to capture the structure of the environment. High prediction errors typically indicate regions in the environment that are novel or unfamiliar to the agent, and low prediction errors signify regions that are more familiar, as the predictor network has learned to approximate the target network's behavior well. Formally, let $f_\theta$ denote the target network with parameters $\theta$, and let $g_\phi$ represent the predictor network with parameters $\phi$. The objective is to minimize the following loss:

$$\mathcal{L}(\phi) = \mathbb{E}_{x \sim \mathcal{X}} \left[ \|g_\phi(x) - f_\theta(x)\|^2 \right], \tag{3}$$

where $\mathcal{X}$ represents the input space (typically the state or observation space of the task). The error between $g_\phi(x)$ and $f_\theta(x)$ reflects how well the predictor network approximates the fixed target network. In environments where the target network is highly unpredictable, the prediction error will be large, indicating the currently visited states are not familiar to the historical states. Utilizing this characteristic, we can assess the familiarity of tasks.

## 3 CURE Method

Sections 3.1 and 3.2 provide a detailed description of the CURE architecture and our approach to estimating task familiarity. Next, Section 3.3 delves into the evaluation of task clarity and the prediction of the success rate.

### 3.1 Method Overview

We propose an overview of our method, CURE, which operates independently of any specific LLM planner. Additionally, this approach is crafted to be plug-and-play, meaning it requires no alterations to the foundational structures of the planners. After task planning is completed, we separately compute epistemic uncertainty and intrinsic uncertainty associated with the planning process. Epistemic uncertainty is further divided into task familiarity and task clarity. First, we employ RND to estimate task familiarity, yielding $A_{\text{sim}}$. Next, task clarity is assessed using two proposed methods: (i) A query-based approach leveraging the LLM, which is relatively slower. (ii) A multi-layer neural network inference approach, which is computationally efficient. Both methods yield a measure of task clarity, denoted as $A_{\text{amb}}$. Finally, intrinsic uncertainty is computed using a multi-layer neural network to infer the expected success rate $p$. The final uncertainty $U$ is determined using the following formula:

$$U = 1 - \alpha_1 \cdot (1 - \alpha_2 \cdot A_{\text{amb}}) \cdot p + \alpha_3 \cdot A_{\text{sim}}, \tag{4}$$

where $\alpha_1, \alpha_2, \alpha_3$ are tunable parameters that balance the contributions of task clarity, expected task success rate, and task similarity to overall uncertainty.

### 3.2 Task Familiarity Assessment

Firstly, we define the two key components in the RND framework: the target network and the predictor network. The outputs of the target network $f_{\text{target}}$ and the predictor network $f_{\text{pred}}$ are the embedding vectors $\mathbf{z}_{\text{target}}$ and $\mathbf{z}_{\text{pred}}$, respectively. Specifically:

$$\begin{aligned} \mathbf{z}_{\text{target}} &= f_{\text{target}}(\mathbf{T}), \\ \mathbf{z}_{\text{pred}} &= f_{\text{pred}}(\mathbf{T}). \end{aligned} \tag{5}$$

where $\mathbf{T}$ denotes the feature vector of the task description, as detailed in Section 2.

During the training phase of the task, we train the RND network by inputting the vectors of known tasks, employing the Mean Squared Error (MSE) as the loss function to optimize the output discrepancy between the target network and the predictor network:

$$\mathcal{L}_{\text{MSE}} = \frac{1}{|\mathcal{D}|} \sum_{(\mathbf{T}, \{\mathbf{S}_i\}) \in \mathcal{D}} \|\mathbf{z}_{\text{target}} - \mathbf{z}_{\text{pred}}\|_2^2, \tag{6}$$

where $\mathcal{D}$ is the dataset of known tasks.

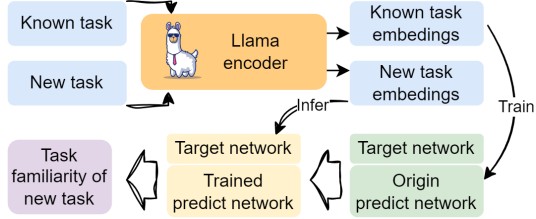

Figure 2: The process of task familiarity assessment

During the inference phase, the new task description vector $\mathbf{T}$ is input into the trained RND network. By computing the Euclidean distance between the outputs of the target network and the predictor network, we obtain the contextual similarity metric $A_{\text{sim}}$:

$$A_{\text{sim}} = \|\mathbf{z}_{\text{target}} - \mathbf{z}_{\text{pred}}\|_2. \tag{7}$$

This similarity metric reflects the degree of similarity between the current task and the known tasks. A larger value of $A_{\text{sim}}$ indicates a greater difference between the current task and known tasks,

suggesting that the task is relatively unfamiliar, thereby appropriately increasing the uncertainty in the output. Conversely, a higher similarity indicates that the current task is more familiar, allowing for a reduction in the uncertainty assessment.

### 3.3 Assessment of Task Clarity and Expected Success Rate

**Slow Evaluation of Task Clarity Using LLM (Ambiguity)** We employed a vanilla approach to assess task clarity by designing a dedicated prompt for ambiguity evaluation. This prompt requires the model to determine whether the given task description provides sufficient information to infer the intended object and location of the user's intent. The specific prompt can be seen in Appendix C:

Through this prompt, we prompt the model to determine whether the task description is sufficiently clear to infer the user's intended object and location. If the model deems the task description inadequate for inference (choose multi item or target location), we consider the task to exhibit semantic ambiguity, thereby increasing the uncertainty of the output. Conversely, if the task description is sufficiently clear (choose one item and one target location), the model's response indicates that the task instructions are adequate, thereby reducing uncertainty.

The final assessment result is obtained as follows:

$$A_{\text{amb}} = \begin{cases} 0 & \text{Sufficient (Information Complete)} \\ 1 & \text{Insufficient (Ambiguity Present)} \end{cases} \tag{8}$$

**Fast Evaluation of Task Clarity and Expected Success Rate Using Uncertainty Assessment Network (UAN)** First, we construct a training dataset using either code generation or LLM-generated data. This dataset comprises the task encoding vector $T$, the presence of task ambiguity $A_{\text{amb}} \in \{0, 1\}$, and whether the planned subtasks were successfully executed $y \in \{0, 1\}$.

We design a multilayer neural network $f_{\text{UAN}}$, which takes as input the feature vector $\mathbf{T}$ and outputs two scalar values:

1. Task clarity score $\hat{A}_{\text{amb}} \in [0, 1]$

2. Expected task success rate $\hat{p} \in [0, 1]$

The network training process employs a cross-entropy loss function:

$$\mathcal{L}_{\text{total}} = \mathcal{L}_{\text{amb}} + \mathbb{I}(A_{\text{amb}} = 0) \cdot \mathcal{L}_{\text{success}} \tag{9}$$

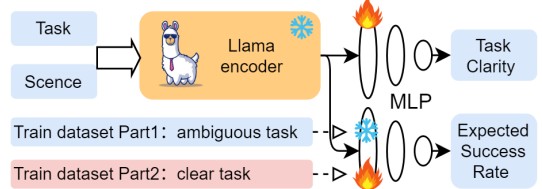

Figure 3: **The process of UAN**. During the training process, Llama consistently remained frozen. For tasks with clear objectives, both task clarity and expected success rate were trained. In tasks with ambiguous goals, only task clarity was trained.

where $\mathbb{I}(\cdot)$ is an indicator function, and the loss terms are defined as follows:

$$\mathcal{L}_{\text{amb}} = - \left( A_{\text{amb}} \log(\hat{A}_{\text{amb}}) + (1 - A_{\text{amb}}) \log(1 - \hat{A}_{\text{amb}}) \right) \tag{10}$$

$$\mathcal{L}_{\text{success}} = - \left( y \log(\hat{p}) + (1 - y) \log(1 - \hat{p}) \right) \tag{11}$$

## 4 Related Work

**LLMs as Planners.** Emergent reasoning enables Large Language Models (LLMs) to decompose tasks into intermediate sub-goals and generate a sequence of planned actions [19]. Through prompting and in-context learning, LLMs can translate natural language human instructions into executable robotic actions based on scene descriptions [20, 21]. RoboCodeX [22], ProgPrompt [23] and CaP [24] decomposes human instructions and leverages code generation to generate actions. RoboMamba [25] adopts a Mamba-based end-to-end vision-language-action (VLA) model for robotic manipulation. RoboMP [26] method enhanced robotic manipulation by integrating multimodal perception and planning to improve generalization and decision-making capabilities. Recent studies further enhance reasoning and planning capabilities by iteratively refining actions via self-reflection during the

planning process [27, 28, 29]. Tool-Planner [30], ReAct [31] and Reflexion [32] focus on *multi-step planning* scenarios, where a robot can execute certain actions, observe state feedback, and re-plan to make corrections. Tree-Planner [33] introduces a tree sampling based solution, whch improvements in error reduction. However, in safety-critical robotic applications, certain invalid actions may lead to irrecoverable and catastrophic safety failures. Therefore, our approach aims to provide an uncertainty estimate after planning, enabling the system to withhold execution of plans with insufficient confidence by anticipating uncertainty before action execution.

**Quantifying Uncertainty in LLMs.** LLMs often experience hallucinations [34], which present two major challenges in planning. First, they might struggle to assess if a plan is achievable given a specific problem description [35, 36]. Second, they can generate inadmissible actions and non-existent objects, requiring translation or expert intervention to correct them [19, 37]. To solve hallucinations, the natural language processing community has shown increasing interest in quantifying uncertainty in the outputs of large language models (LLMs) [38, 39, 40, 41], calibrating this uncertainty empirically for accuracy [42, 43, 44], and examining the reliability of models [45, 46]. Some researchers have also focused on quantifying uncertainty in LLMs within the domain of robotic task planning. For instance, the KnowNo framework [47] treats task-level planning as a multiple-choice question answering (MCQA) problem, and uses conform prediction to output a subset of candidate plans generated by the LLM. IntroPlan [48], aligns the robot's uncertainty with the intrinsic ambiguity of task specifications before predicting a high-confidence subset of plans. However, these studies primarily focus on the success rate of the overall task, with less attention given to the quality of uncertainty estimation, and the distinct sources of uncertainty were overlooked. This paper presents a supervised learning method using neural network outputs to estimate uncertainty, distinguishing between epistemic uncertainty and intrinsic uncertainty. This approach offers a more comprehensive understanding and capture of uncertainty in task planning. It provides a more accurate and faster estimation of task planning uncertainty compared to existing methods.

# 5   Experiments and Result

We evaluate our methods across a diverse range of language-instructed tasks and environments, demonstrating their effectiveness in generating uncertainty assessments that are strongly correlated with success rates.

**Baselines.**   We consider 8 baselines, including KnowNo [47], IntroPlan [48], and several uncertainty estimation approaches in large language models (LLMs) [38]. **KnowNo** [47] treats task-level planning as a multiple-choice question answering (MCQA) problem and employs conformal prediction to output a subset of candidate plans generated by the LLM. Since there is no uncertainty output value, the softmax probability of the best option is directly used as the uncertainty estimate for task planning. **IntroPlan** [48] introduces a knowledge base search process, incorporating the retrieved knowledge into the context when estimating uncertainty to enhance prediction accuracy. **Multi-step** [38] decomposes the reasoning process into steps and extracts confidence levels for each step, which can help mitigate overconfidence. **Top-k** [38] prompts the model to output multiple task plans and their respective confidence levels. **CoT** [38] uses zero-shot chain-of-thought (CoT), which has been proven effective in inducing reasoning processes and improving model accuracy across various datasets. **Vanilla** [38] directly instructs the LLM to state its confidence. **Self-probing** [38] first generates an answer and then retrieves the confidence expressed verbally in another independent chat session. **Self-probing-log** is similar to self-probing but applies softmax to the probabilities of outputting "yes" and "no," using the probability of "yes" as the LLM's confidence.

**Proposed Methods (Specific introduction provided in Section 3.3 and Appendix D).**   **Ambiguity** An inquiry is made into whether the directive to the LLM is ambiguous. If it is ambiguous, the confidence in the question is assessed as 0; otherwise, it is assessed as 1. **CURE-Ambiguity** The confidence value is derived based on the value of ambiguity, in conjunction with the output of CURE. **KnowNo-Ambiguity** The confidence value is assessed based on the value of ambiguity, in combination with the output of KnowNo methodologies. **CURE w/o sim** Confidence is solely derived from the output of the UAN network. **CURE** The confidence value is derived from Equation (15).

**Metrics.** We introduces a novel metric, termed **SR-HR-AUC**, to more fairly assess the performance of uncertainty estimation methods. The metric quantifies the accuracy of uncertainty estimation by

analyzing the variation in the success rate (SR) of a task at different help rates (HR). In addition to the **SR-HR-AUC** metric, we also utilize the **Spearman's rank correlation coefficient** to evaluate the relationship between uncertainty estimation and task success rate. To further assess whether the correlation is significant, we compute the corresponding **p-value**. Both the Spearman coefficient and the associated p-value together provide a quantitative evaluation of whether a significant monotonic relationship exists between uncertainty estimation and task success rate, offering additional statistical support for the validity of the uncertainty estimation method. A detailed explanation of these metrics can be found in Appendix A.

**Implementation Details.** To facilitate reproducibility, we adopt open-source large language models (LLMs) as the planner in our approach. For the kitchen operation experiments, we utilize `Llama-3.3-70B-Instruct` to accomplish the tasks. In the case of the rearrangement experiments, which are relatively simpler, we employ `Llama-3.2-8B-Instruct` to complete the tasks. In this paper, we set $\alpha_1 = 1$, $\alpha_2 = 0.6$, and $\alpha_3 = 30$. Appendix E contains the hyperparameter search experiments for these parameters. All experiments were conducted on a computing server equipped with dual Intel Xeon Gold 6348 processors , 512GB of RAM, and four NVIDIA A100-PCIE-40GB GPUs. Conducting the experiment took approximately 12 hours.

## 5.1 Mobile Manipulator in a Kitchen

Table 1: Results for Mobile Manipulator in a Kitchen

| experiment name | spearman | p-value | SR-HR-AUC |
|---|---|---|---|
| introplan | 0.030 | 0.599 929 | 0.005 |
| self-probing-log | 0.181 | 0.001 651 | 0.094 |
| self-probing | 0.111 | 0.054 750 | 0.120 |
| vanilla | 0.221 | 0.000 115 | 0.236 |
| cot | 0.247 | $1.467 \times 10^{-5}$ | 0.273 |
| top-k | 0.254 | $8.286 \times 10^{-6}$ | 0.286 |
| multi-step | 0.247 | $1.520 \times 10^{-5}$ | 0.297 |
| KnowNo | 0.336 | $2.441 \times 10^{-9}$ | 0.395 |
| **Ambiguity(Ours)** | 0.433 | $3.802 \times 10^{-15}$ | 0.371 |
| **CURE w/o sim(Ours)** | 0.417 | $4.431 \times 10^{-14}$ | 0.483 |
| **KnowNo-Ambiguity(Ours)** | 0.426 | $1.114 \times 10^{-14}$ | 0.501 |
| **CURE(Ours)** | 0.454 | $1.079 \times 10^{-16}$ | 0.534 |
| **CURE-Ambiguity(Ours)** | **0.466** | $1.460 \times 10^{-17}$ | **0.547** |

For this environment, we adopt the task specifications defined in KnowNo [47]. Each scenario involves a mobile robot positioned in front of a kitchen counter, adjacent to a set of recycling/compost/trash bins. The task entails picking up objects from the counter and placing them either into one of the bins or elsewhere on the counter. The environment exhibits certain ambiguities, including scenarios involving potentially unsafe actions.

The results presented in Table 1 offer a comprehensive evaluation of various uncertainty estimation methodologies applied to the mobile manipulator task in a kitchen environment.

The baseline methods demonstrate varying degrees of effectiveness in estimating uncertainty and predicting task success, as evidenced by both Spearman correlation and SR-HR-AUC metrics. IntroPlan shows a relatively low Spearman correlation of 0.030 (p = 0.60) and the lowest SR-HR-AUC value of 0.005, possibly due to its high initial success rate. Self-probing-log achieves a Spearman coefficient of 0.181 (p = 0.0017) and an SR-HR-AUC value of 0.094. The vanilla approach yields a higher correlation of 0.221 (p = 0.000115) and an SR-HR-AUC value of 0.236. CoT performs slightly better, with a Spearman coefficient of 0.247 (p = 1.467e-5) and an SR-HR-AUC value of 0.273.Top-k and Multi-step methods exhibit the highest correlations among the baseline methods—0.254 and 0.247, respectively—and corresponding SR-HR-AUC values of 0.286 and 0.297. Notably, KnowNo outperforms the other baselines with a correlation coefficient of 0.336 (p = 2.441e-9) and an SR-HR-AUC value of 0.395, highlighting its superior performance in this context.

The proposed methods significantly outperform the baseline approaches. The Ambiguity method achieves a Spearman correlation of 0.433 (p-value: 3.802e-15), confirming the importance of ambiguity recognition in uncertainty estimation. The CURE w/o sim and KnowNo-Ambiguity methods further enhance performance, with correlations of 0.417 and 0.426, respectively, both showing highly significant p-values. The CURE method achieves an even higher correlation of 0.454 (p-value: 1.079e-16), demonstrating the benefits of integrating UAN and RND approaches. The CURE-Ambiguity method emerges as the most effective, achieving the highest Spearman correlation of 0.466 (p-value: 1.460e-17) and the best SR-HR-AUC value of 0.547.

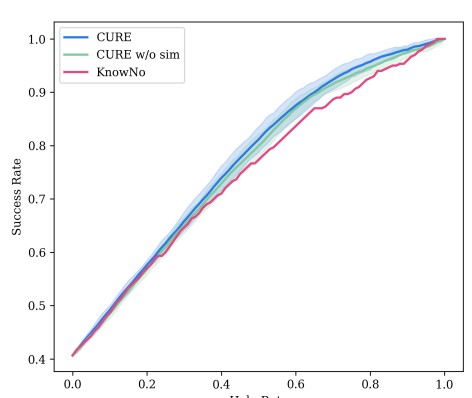

The help-success curves of these methods are shown in Figures 4 and 6. It can also be observed from the figures that the curve of the proposed method has a larger area. As observed from Figure 4, the confidence interval of the proposed method is notably narrow, indicating that the performance of CURE is highly stable.

In conclusion, the proposed methods consistently outperform the baseline approaches, providing more accurate and reliable uncertainty estimates in the kitchen operation task. These results validate the novel methodologies and metrics introduced in this study, offering significant advancements in uncertainty estimation for LLM-instructed tasks.

Figure 4: **the Help Rate-Success Rate Curve of CURE and KnowNo for Mobile Manipulator in a Kitchen** two variants of the CURE algorithm were executed 11 times. The lighter-colored region represents the $2\sigma$ confidence interval.

### 5.1.1 Result for CURE+IntroPlan

We conducted additional experiments to evaluate the Overstep Rate, Overask Rate, and Help Rate of the IntroPlan method, CURE method, and IntroPlan + CURE method when the target success rate is 90%. The experimental results are shown in Table 2.

Table 2: Comparison of Overstep Rate, Help Rate and Overask Rate for CURE and IntroPlan.

| Metric | CURE | IntroPlan + CURE | IntroPlan |
|---|---|---|---|
| **Overstep Rate** ↓ | 34.88% | **21.58%** | 31.82% |
| **Help Rate** ↓ | 71.33% | 53.67% | **19.33%** |
| **Overask Rate** ↓ | **29.44%** | 53.42% | 51.72% |

The results show that the IntroPlan + CURE method achieved the absolute optimal effect in terms of Overstep Rate, significantly reducing unsafe incidents. In terms of Help Rate, IntroPlan demonstrated the lowest rate, indicating higher confidence and reduced need for human intervention. However, considering the small difference between IntroPlan + CURE and IntroPlan in the Overask Rate metric, it suggests that the help behavior of IntroPlan + CURE was not ineffective, significantly improving safety performance. CURE performed best in the Overask Rate, partly due to the lower initial success rate of its base method, KnowNo, making the effectiveness of the help more significant.

### 5.2 Tabletop Rearrangement

In this task, the robot arm is required to rearrange objects on a table within the PyBullet simulator. Each scene is initialized with three bowls and three blocks, each of which is colored blue, green, and yellow, respectively. The task requires the robot to move a specific number of blocks or bowls to a designated position relative to another object. For example, "Move the green block to the left of the blue bowl."

In the tabletop rearrangement task, we use the task definitions from KnowNow and conduct experiments within the PyBullet simulation environment. The robot is required to parse ambiguous user instructions and perform the appropriate object rearrangement actions. The primary goal of the experiment is to evaluate the effectiveness of different uncertainty estimation methods in handling

Table 3: Results for Tabletop Rearrangement

| experiment name | spearman | p-value | SR-HR-AUC |
|---|---|---|---|
| vanilla | 0.328 | $6.519 \times 10^{-9}$ | 0.293 |
| cot | 0.312 | $3.901 \times 10^{-8}$ | 0.320 |
| self-probing | 0.282 | $7.751 \times 10^{-7}$ | 0.301 |
| multi-step | 0.161 | $5.222 \times 10^{-3}$ | 0.192 |
| top-k | 0.155 | $7.281 \times 10^{-3}$ | 0.146 |
| KnowNo | 0.302 | $9.609 \times 10^{-8}$ | 0.351 |
| **CURE w/o sim(Ours)** | 0.610 | $6.347 \times 10^{-32}$ | 0.703 |
| **CURE(Ours)** | **0.635** | $2.827 \times 10^{-35}$ | **0.732** |

ambiguous instructions. Due to the simplicity of the experiment and the effectiveness of the fast methods, we did not test the slower methods(LLM based Ambiguity method).

The results presented in Table 3 offer a comparative analysis of the baseline and proposed methods in estimating uncertainty during object rearrangement tasks. Among baseline methods, the Vanilla approach achieves a Spearman correlation of 0.328 (p = 6.519e-9), outperforming other baselines such as CoT (0.312, p = 3.901e-8) and Self-probing (0.282, p = 7.751e-7). This suggests that direct confidence elicitation from LLMs can provide moderately reliable uncertainty estimates for simpler rearrangement tasks. However, decomposition-based methods like Multi-step (Spearman = 0.161) and Top-k (Spearman = 0.155) exhibit notably weaker correlations, indicating that stepwise confidence aggregation may not generalize well to this domain. KnowNo, while achieving a competitive Spearman coefficient of 0.302 (p = 9.609e-8), remains limited compared to our proposed methods, highlighting inherent constraints of KnowNo approaches in handling instruction ambiguities.

The proposed methods exhibit substantial improvements. CURE w/o sim achieves a Spearman correlation of 0.610 (p = 6.347e-32), nearly doubling the performance of the best baseline. This demonstrates the effectiveness of the UAN in capturing task-specific uncertainties. The integration of RND in CURE further enhances performance, yielding the highest Spearman coefficient of 0.635 (p = 2.827e-35) and SR-HR-AUC of 0.732. The extreme statistical significance (p < 1e-30) of both methods underscores the robustness of their uncertainty estimates.

The SR-HR-AUC metric reveals practical implications: CURE's AUC of 0.732 indicates superior ability to modulate help requests based on uncertainty, compared to KnowNo's 0.351. This 108% improvement demonstrates that our methods more effectively align uncertainty estimates with actual task success probabilities, critical for safe human-robot collaboration.

To further visually demonstrate the performance of different methods, we plotted the help rate-success rate curves for the baseline method and the proposed method (Figures 7). As shown in the figures, the area under the curve for the CURE method is significantly larger than that of the baseline method, indicating its superior ability to identify tasks with high uncertainty and take appropriate actions.

In summary, the proposed CURE methodologies markedly outperform the baseline methods, providing more accurate and reliable uncertainty estimates in the tabletop rearrangement task.

## 6 Conclusion

This paper presents CURE, an innovative framework designed to enhance reliable planning within robotics applications through the utilization of LLMs. CURE significantly improves the alignment between uncertainty estimates and actual execution outcomes. Empirical evaluations conducted on tasks involving kitchen manipulation and tabletop rearrangement demonstrate that CURE surpasses existing methodologies, offering uncertainty estimates that exhibit a stronger correlation with the rates of task success. Furthermore, CURE is characterized by its ease of integration with any LLM-based planner, thereby requiring minimal effort for implementation.

**Limitations and Future Work.** A primary limitation of our current work is that the predictive model necessitates pretraining on existing datasets, and the model lacks generalizability, requiring deployment tailored to specific task sets. Secondly, the current generation of uncertainty has not

been calibrated with actual success rates, necessitating an additional calibration step to align with the confidence in execution.

Future research endeavors will focus on refining the CURE framework by integrating dynamic task familiarity models capable of real-time adaptation, thereby augmenting the framework's robustness across a broader array of increasingly complex task scenarios. Additionally, the incorporation of physical concepts into the uncertainty prediction model is proposed as a means to enhance the model's generalizability, allowing it to better accommodate a wider variety of tasks and conditions.

## 7 Broader Impact

In this article, we propose a fine-grained method for uncertainty estimation by quantifying the inference confidence of a foundational model through both the model's epistemic uncertainty and the intrinsic uncertainty of the environment. CURE operates as a plug-and-play approach, which requires no modification of existing workflows, and can be seamlessly integrated into current LLM-based robotic planners as a supplement. Furthermore, we contend that CURE can serve as a general method, extending the inference capabilities of foundational models beyond robotic applications.

However, as previously mentioned, during the implementation of our model, our approach continues to yield high uncertainty in unseen environments. This maintains safety but necessitates new training to improve accuracy. This limitation in generalization performance restricts the applicability of CURE.

**Acknowledgments and Disclosure of Funding**    This work is supported by the National Key Research and Development Program of China (Grant No.2024YFE0210900), the National Natural Science Foundation of China (Grant No.62306242), the Young Elite Scientists Sponsorship Program by CAST (Grant No. 2024QNRC001), the Yangfan Project of the Shanghai (Grant No.23YF11462200) and the Key Scientific Research Project Plan for Higher Education Institutions in Henan Province (Grant No.26A520006).

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

# A  Matrics Detail

## A.1  SR-HR-AUC

**SR-HR-AUC** offers a more equitable evaluation of the performance of uncertainty estimation methods. This metric quantifies the accuracy of uncertainty estimation by analyzing the variation in the success rate (SR) at different help rates (HR). Specifically, in a given task, a robot can ensure the success of the task by requesting human assistance; if no assistance is requested, the task success rate is determined by the success rate of the planning model based on a large language model (LLM). Assume there are 100 tasks, each planned using an LLM, and uncertainty is evaluated using an uncertainty estimation module. At various help rates, tasks with the highest uncertainty are selected for human intervention, while the remaining tasks are autonomously executed by the robot based on the LLM's plan, resulting in an overall success rate.

At a help rate of 0, the task success rate is $y$, which reflects the actual performance of the planning model; at a help rate of 1, the task success rate is necessarily 100%. By plotting the curve of help rate versus success rate, the area under the curve (AUC) can be used to assess the precision of uncertainty estimation. However, to eliminate the influence of the model's planning performance on uncertainty estimation, this study introduces an additional normalization procedure.

Specifically, two benchmark curves are defined in this study:

1. **Random Evaluation Curve**: Assumes that the uncertainty of all tasks is a random value. This curve linearly ascends from the baseline success rate point $(0, y)$ to $(1, 1)$.

2. **Perfect Uncertainty Evaluation Curve**: Indicates a strong correlation between uncertainty and task success rate. It ascends from $(0, y)$ to $(1 - y, 1)$, then remains constant.

where $y$ is the task success rate when help rate is 0.

By calculating the area difference between the actual uncertainty evaluation curve and the random evaluation curve, and dividing this by the area difference between the perfect evaluation curve and the random evaluation curve, a normalized AUC value is obtained. This normalization process effectively eliminates the influence of the baseline success rate, allowing SR-HR-AUC to more accurately reflect the quality of uncertainty estimation.

As shown in the figure, the actual uncertainty evaluation curve (orange curve) lies between the random evaluation curve (purple dashed line) and the perfect evaluation curve (green solid line). By calculating the areas under each curve and applying the normalization formula described above, the resulting SR-HR-AUC value provides a reliable measure of whether uncertainty estimation is closely related to task success rate, thus offering a robust evaluation of uncertainty estimation methods.

$$\text{SR-HR-AUC} = \text{Normalized AUC} = \frac{\text{Actual AUC} - \text{Random AUC}}{\text{Perfect AUC} - \text{Random AUC}}$$

## A.2  Spearman's Rank Correlation Coefficient

Spearman's rank correlation coefficient is a non-parametric statistical method used to assess the monotonic relationship between two variables. Unlike Pearson's correlation coefficient, Spearman's coefficient does not require the data to follow a normal distribution or exhibit a linear relationship, making it suitable for various data types, particularly when the data does not conform to a linear relationship, thus demonstrating better robustness.

Specifically, the Spearman coefficient evaluates the relationship between variables by ranking them and then calculating the correlation between these ranks. Its value ranges from $[-1, 1]$, where:

- A value of **1** indicates a perfect positive correlation between the two variables (i.e., as one variable increases, the other variable monotonically increases);

- A value of **-1** indicates a perfect negative correlation between the two variables (i.e., as one variable increases, the other variable monotonically decreases);

- A value of **0** indicates no monotonic relationship between the two variables.

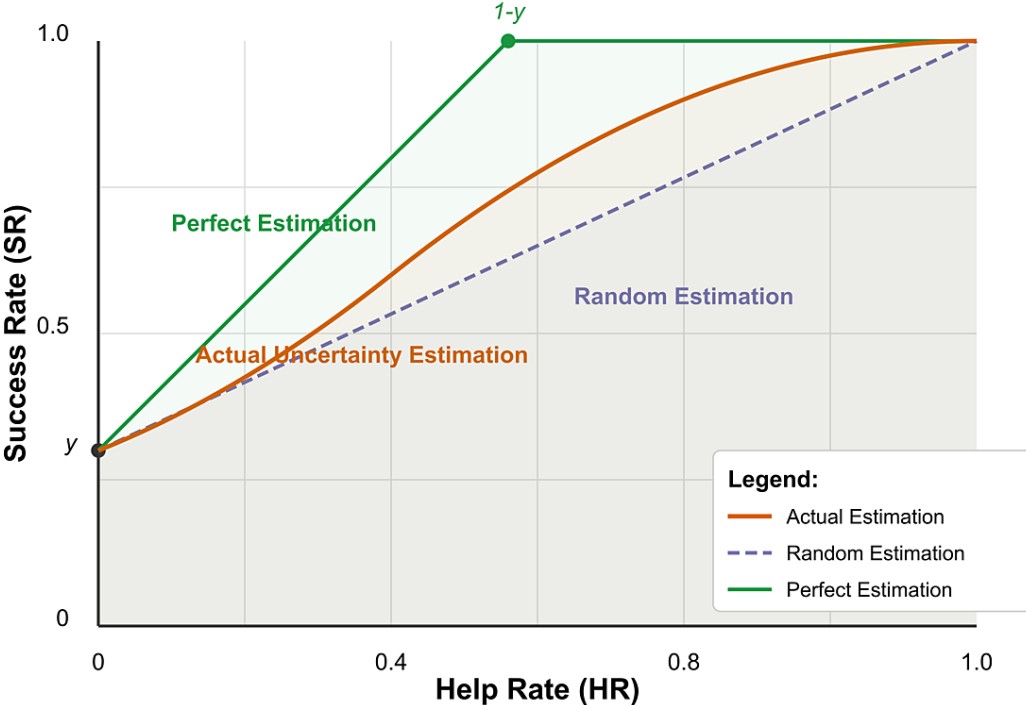

Figure 5: The SR-HR-AUC metric is used to evaluate the performance of uncertainty estimation. The figure shows three curves: the actual uncertainty evaluation curve (orange solid line), the random evaluation curve (purple dashed line), and the perfect uncertainty evaluation curve (green solid line). The horizontal axis represents the help rate (HR), and the vertical axis represents the success rate (SR). By calculating the area difference between the actual curve and the random curve, and dividing by the area difference between the perfect curve and the random curve, a normalized AUC value is obtained. This metric effectively eliminates the influence of the baseline success rate, providing a more accurate reflection of the quality of uncertainty estimation.

To further assess the significance of the correlation, we compute the corresponding **p-value**. The p-value is used to measure the consistency between the observed result and the null hypothesis (usually "no correlation"). A small p-value (typically less than 0.05) indicates that the null hypothesis is rejected, suggesting that the correlation between the two variables is statistically significant, meaning there is a significant monotonic relationship. Conversely, a larger p-value indicates that the null hypothesis cannot be rejected, suggesting that the correlation is not significant.

Therefore, the Spearman coefficient, along with the corresponding p-value, provides a quantitative evaluation of whether a significant monotonic relationship exists between uncertainty estimation and task success rate, offering further statistical support for the validity of uncertainty evaluation methods.

## B   Additional Results

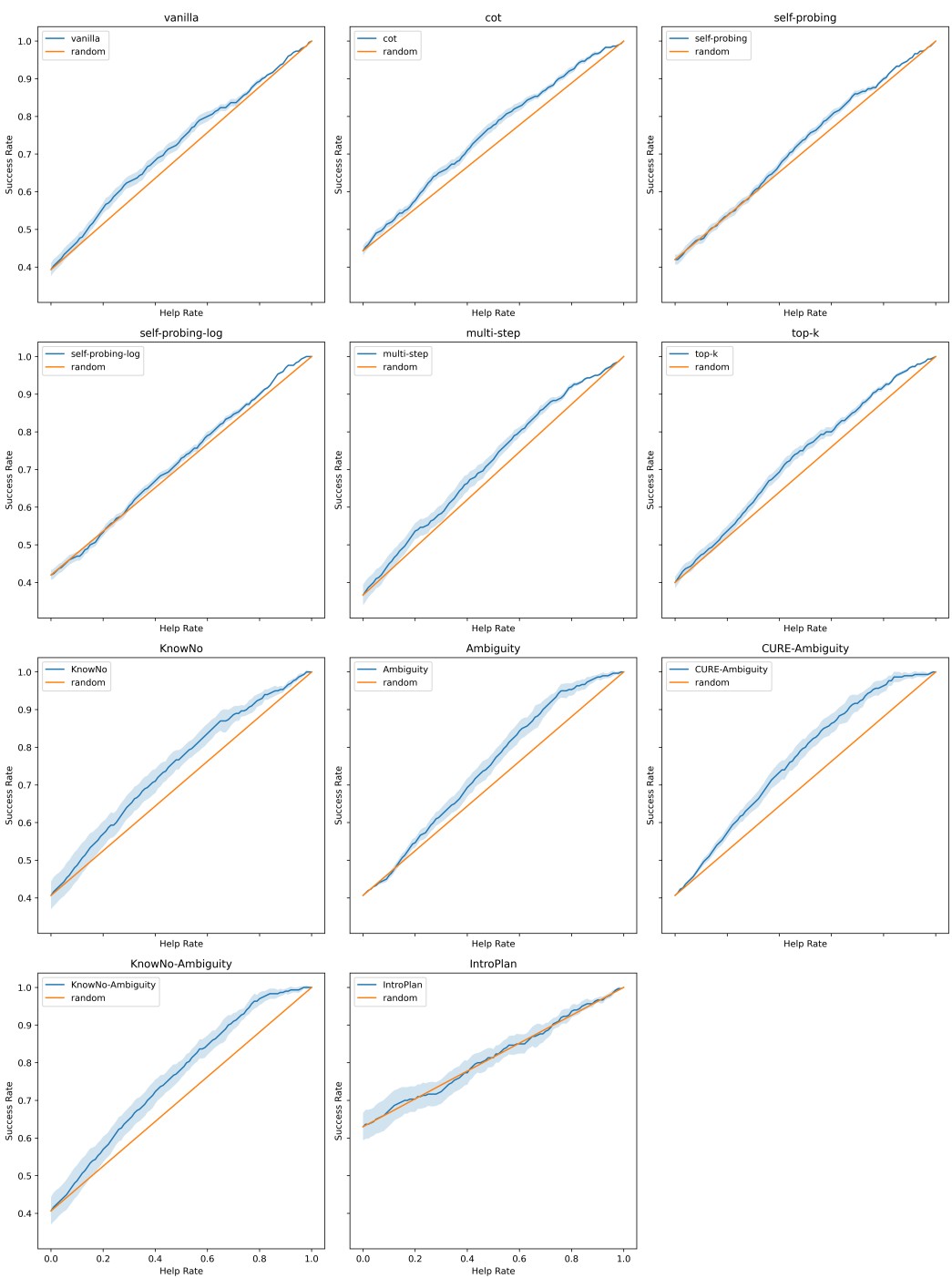

Figure 6: The Help Rate-Success Rate Curve for Mobile Manipulator in a Kitchen

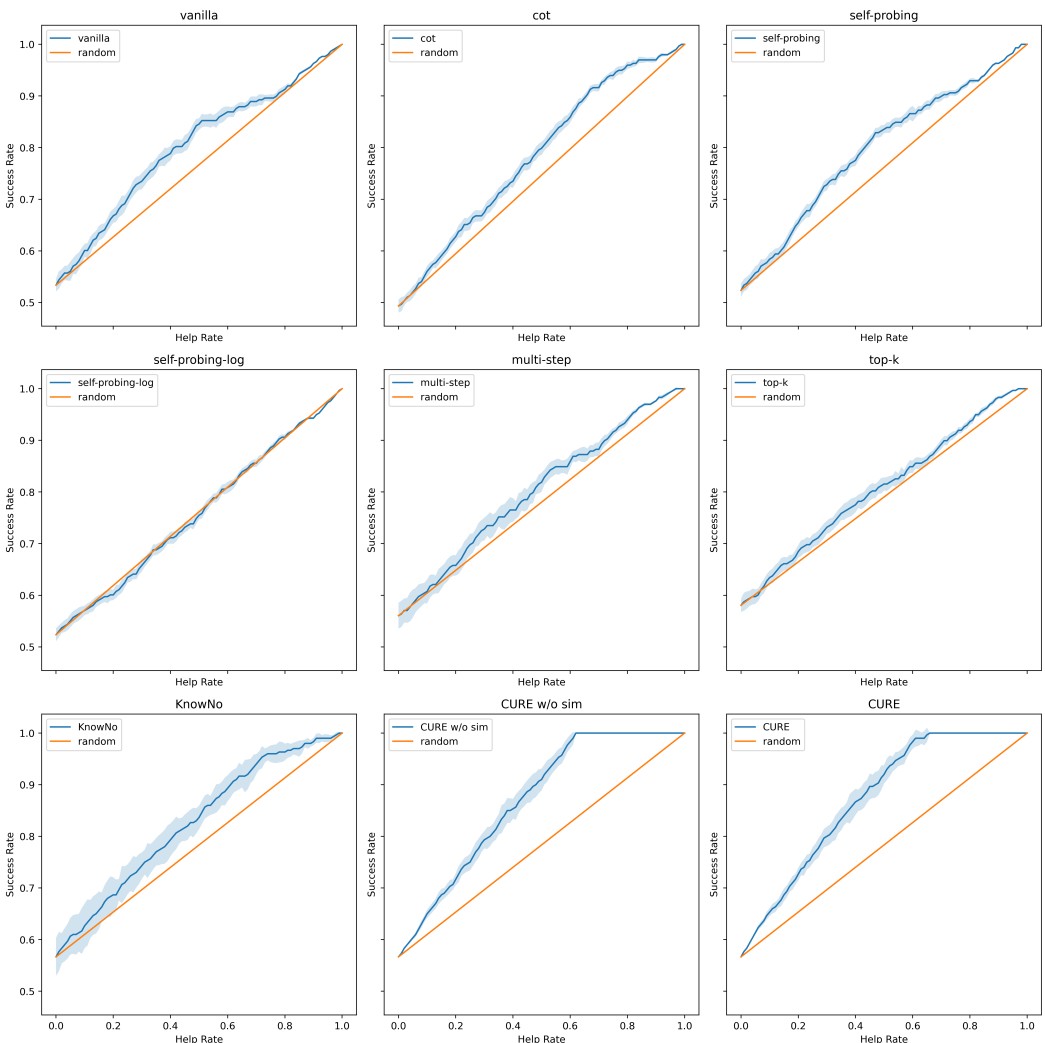

Figure 7: The Help Rate-Success Rate Curve For Tabletop Rearrangement

### B.1 Dataset scale result

We conducted experiments on dataset scale, and the results are as follows: The experimental results

| Training Set Size | 100 | 300 | 1000 | 3000 | 10000 | 30000 | 100000 |
|---|---|---|---|---|---|---|---|
| SR-HR-AUC | 0.125 | 0.146 | 0.174 | 0.234 | 0.362 | 0.430 | 0.547 |

Table 4: Experimental results on dataset scale.

demonstrate that increasing the dataset scale significantly improves the performance of the CURE model. When the dataset reaches about 10,000 samples, the performance improvement becomes noticeable, and further scaling up to 100,000 samples still yields continuous gains.

### B.2 Calibrate result

Our approach involved splitting the test set into two parts: one for calibration and the other for testing. Given a target success rate, we first determined a threshold on the calibration set. Tasks with confidence below the threshold were completed with human assistance, while tasks with confidence above the threshold were executed autonomously. By adjusting the threshold, we could make the success rate approach the target success rate, thereby achieving calibration.

We tested this method for a target success rate of 85%. The results showed that the success rate on the test set reached 84.33%, differing from the target by only 0.67%. This demonstrates that the CURE method achieves good calibration accuracy.

## C  Prompt Template

In the following prompt, "SCENE" refers to the objects present in the current environment, for example, [Coca-Cola, Sprite, apple]. "TASK" refers to the current instruction, such as [give me something to drink].

### C.1 Ambiguity method prompt

You are a robot in kitchen. You are given a scenario and a task. You need to place an item in a new location.
Scenario: "SCENE"
Task: "TASK"
Optional items are: "SCENE"
Optional target locations are: "SCENE", user's hand, top drawer, bottom drawer, garbage can.
Please tell me what items you choose from and where is the target locations. You can choose one or more. You need to choose every item and target location fit the task.
(Based on common-sense reasoning, extreme special cases should be disregarded.)

### C.2 Vanilla prompt

You are a robot operating in an office kitchen.
You are in front of a counter with two closed drawers, a top one and a bottom one.
There is also a landfill bin, a recycling bin, and a compost bin.
On the counter, there is "SCENE"
"TASK"
provide what will you do and your confidence in this answer.
Note: The confidence indicates how likely you think what will you do is true.
Note: The action you take should be one of [pick-up, bring, move, open, close] and something you seen(should be a single full name) [to where(should be a single full name) you seen if needed].
Use the following format to answer:
"Action:[What will you do]

Confidence: [Your confidence level, please only include the numerical number in the range of 0-100]%"
Only the answer and confidence, don't give me the explanation.
Now, please answer this question and provide your confidence level.

### C.3   COT prompt

You are a robot operating in an office kitchen.
You are in front of a counter with two closed drawers, a top one and a bottom one.
There is also a landfill bin, a recycling bin, and a compost bin.
On the counter, there is "SCENE"
"TASK"
analyze step by step, provide what will you do and your confidence in this answer.
Note: The confidence indicates how likely you think your action is true. Use the following format to answer:
"Explanation: [insert step-by-step analysis here]
Action:[What will you do Here]
Confidence:[Your confidence level, please only include the numerical number in the range of 0-100]%"
Note: The action you take should be one of [pick-up, bring, move, open, close] and something you seen(should be a single full name) [to where(should be a single full name) you seen if needed].
Only give me the reply according to this format, don't give me any other words.
Now, please answer this question and provide your confidence level. Let's think it step by step.

### C.4   Self probing prompt

You are a robot operating in an office kitchen.
You are in front of a counter with two closed drawers, a top one and a bottom one.
There is also a landfill bin, a recycling bin, and a compost bin.
On the counter, there is "SCENE"
"TASK"
What will you do?
Note: The action you take should be one of [pick-up, bring, move, open, close] and something you seen(should be a single full name) [to where(should be a single full name) you seen if needed].

"LLM response"

According to your response, what is your confidence in this action? reply with the confidence only. the confidence should be a percentage.

### C.5   Multi step prompt

You are a robot operating in an office kitchen.
You are in front of a counter with two closed drawers, a top one and a bottom one.
There is also a landfill bin, a recycling bin, and a compost bin.
On the counter, there is "SCENE"
"TASK"
What will you do? only interact with the objects in the scene.
Read the question, break down the problem into K steps, think step by step,
give your confidence in each step, and then derive your final answer and your confidence in this answer.
Note: The confidence indicates how likely you think your answer is true.
Use the following format to answer:
"Step 1: [Your reasoning], Confidence: [ONLY the confidence value that this step is correct]...
Step K: [Your reasoning], Confidence: [ONLY the confidence value that this step is correct]Final Answer and Overall Confidence (0-100): [ONLY the answer type; not a complete sentence], [Your confidence value]%"

### C.6 Top-k prompt

You are a robot operating in an office kitchen.
You are in front of a counter with two closed drawers, a top one and a bottom one.
There is also a landfill bin, a recycling bin, and a compost bin.
On the counter, there is "SCENE"
"TASK"
What will you do? only interact with the objects in the scene.
Provide your k best guesses and the probability that each is correct (0% to 100%) for the following question.
Give ONLY the task output description of your guesses and probabilities, no other words or explanation.
Note: The action you take should be one of [pick-up, bring, move, open, close] and something you seen(should be a single full name) [to where(should be a single full name) you seen if needed].
For example:
G1: <ONLY the action description of first most likely guess; not a complete sentence, just the guess!>
P1: <ONLY the probability that G1 is correct, without any extra commentary whatsoever; just the probability!>
...
Gk: <ONLY the action description of k-th most likely guess>
Pk: <ONLY the probability that Gk is correct, without any extra commentary whatsoever; just the probability!>

## D    Additional implementation details

**CURE-Ambiguity**

$$U = 1 - \alpha_1 \cdot (1 - \alpha_2 \cdot 0.5 \cdot (A_{\text{amb1}} + A_{\text{amb2}})) \cdot p + \alpha_3 \cdot A_{\text{sim}} \tag{12}$$

Where $A_{\text{amb1}}$ and $A_{\text{amb2}}$ are derived from Ambiguity and UAN, respectively.

**Ambiguity**

$$U = 1 - \alpha_1 \cdot (1 - \alpha_2 \cdot A_{\text{amb1}}) \cdot p + \alpha_3 \cdot A_{\text{sim}} \tag{13}$$

Where $A_{\text{amb1}}$ is derived from Ambiguity.

**KnowNo-Ambiguity**

$$U = 1 - \alpha_1 \cdot (1 - 0.5 \cdot A_{\text{amb1}}) \cdot p \cdot C_{\text{KnowNo}} \tag{14}$$

Where $C_{\text{KnowNo}}$ is the confidence from KnowNo and $A_{\text{amb1}}$ is derived from Ambiguity.

**Complete Procedure for Kitchen Operation Experiment**

First, we grasp the initial information including the scene and task, for example:

- **Scene:** a bottled unsweetened tea, an orange, and a bag of jalapeno chips
- **Task:** Put jalapeno chips in the drawer.

Next, we supplement the complete scene information to form a complete scene description:

We: You are a robot operating in an office kitchen. You are in front of a counter with two closed
We: On the counter, there is an orange soda, a Pepsi, and an apple.

Based on this, we generate operation instructions according to the complete scene and task. This generation process is consistent with KnowNo[**?** ]. It is emphasized that this generation method is independent of the method proposed in this paper and can adopt any method to generate operation

instructions. The generated operation instruction is (if the instruction is not in this format, it will be reformatted by LLM):

```
action: pick-up jalapeno chips to bottom drawer
```

Next, we combine the aforementioned content into a prompt with specific content:

```
You are a human and there is a robot operating in an office kitchen. The robot is in front of a c
On the counter, there is {scene}.
You say: "{task}".
Then the robot {action}.
```

We input this prompt into the Llama3-8B model and extract the last layer's hidden state (`last_hidden_state`) as the feature vector $T$.

Subsequently, the feature vector is input into the UAN network and RND network to obtain the final expected success rate $p$, ambiguity $A_{\text{amb}}$, and familiarity $A_{\text{sim}}$:

$$p = 0.962$$
$$A_{\text{amb}} = 0.928$$
$$A_{\text{sim}} = 0.048$$

Finally, the uncertainty (Uncertainty) is calculated using Eq.4; Confidence is calculated as: $1 - U$. The confidence value in the example is $0.378$.

## E  Hyperparameter Search Experiments

In this section, we detail the hyperparameter search experiments conducted to fine-tune the parameters $\alpha_1 = 1$, $\alpha_2 = 0.6$, and $\alpha_3 = 30$ in the uncertainty estimation formula, as shown in Equation 15. These experiments were performed within the context of the Mobile Manipulator in a Kitchen environment.

$$U = 1 - \alpha_1 \cdot (1 - \alpha_2 \cdot A_{\text{amb}}) \cdot p + \alpha_3 \cdot A_{\text{sim}}, \tag{15}$$

$\alpha_1$ was set to 1 and remained unchanged throughout the experiments. This decision was made to simplify the equation's structure, ensuring that the primary scaling of the output confidence is directly influenced by the product of the other components. By maintaining $\alpha_1$ at a constant value, we ensure that the adjustments in confidence primarily reflect changes in ambiguity ($A_{\text{amb}}$) and similarity ($A_{\text{sim}}$), thereby focusing the exploration on the interactions of these factors. The hyperparameter search aimed to optimize $\alpha_2$ and $\alpha_3$ to balance the contributions of ambiguity and similarity to the overall uncertainty score. Several values were tested for each parameter to assess their impact on the performance metrics, particularly the Spearman correlation coefficient and SR-HR-AUC.

$\alpha_2$ **Search:**  The parameter $\alpha_2$ modulates the influence of ambiguity on the uncertainty score. We explored values ranging from 0.1 to 0.9 while keeping $\alpha_3 = 30$, with the corresponding SR-HR-AUC values depicted in Figure 8. The experiments indicated that lower values resulted in diminished sensitivity to the ambiguity component, while higher values led to excessive influence, sometimes overshadowing the task success probability ($p$). The optimal value of $\alpha_2 = 0.6$ was chosen as it provided a balanced contribution to the uncertainty score, resulting in improved correlation and SR-HR-AUC metrics.

$\alpha_3$ **Search:**  The parameter $\alpha_3$ adjusts the weight of the similarity component. Values from 0 to 3000 were tested while maintaining $\alpha_2 = 0.6$, as shown in Figure 8. Findings showed that lower values inadequately captured similarity nuances, whereas higher values introduced excessive noise into the estimation process. $\alpha_3 = 30$ emerged as the optimal setting, aligning similarity's contribution with the overall uncertainty, thereby maximizing the predictive performance as measured by the evaluation metrics. Overall, these hyperparameter settings were determined through exhaustive trials aimed at enhancing the model's ability to accurately estimate uncertainty and predict task success. The chosen values of $\alpha_2 = 0.6$ and $\alpha_3 = 30$ reflect the optimal balance between the distinct components of ambiguity and similarity, leveraging their individual strengths to achieve robust uncertainty estimation.

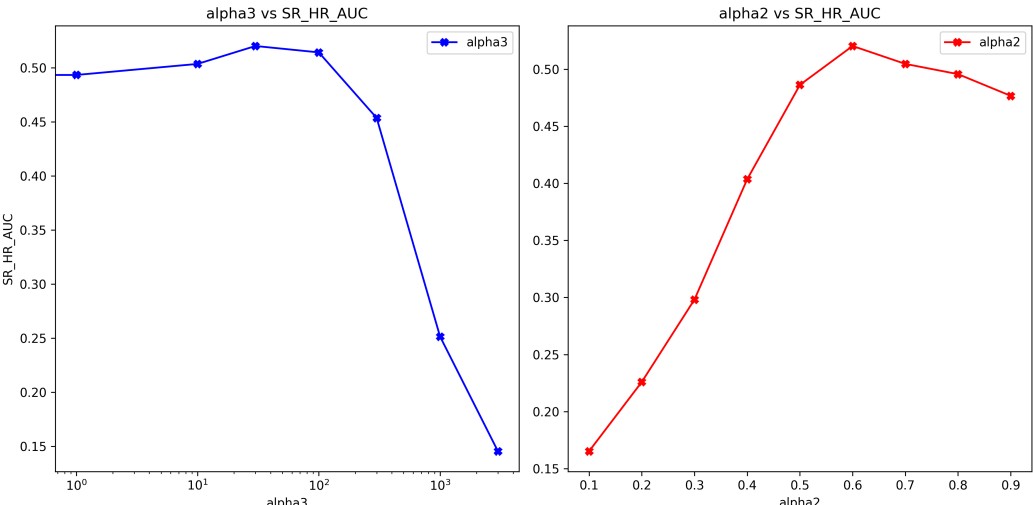

Figure 8: SR-HR-AUC values for different settings of $\alpha_2$ and $\alpha_3$.

