# OpenReview forum: "Towards Reliable LLM-based Robots Planning via Combined Uncertainty Estimation"
_NeurIPS.cc/2025/Conference — NeurIPS 2025 poster_

### Official Review · Reviewer_hD99 · 2025-06-03

**Clarity:** 2
**Significance:** 2
**Originality:** 3
**Rating:** 4
**Confidence:** 3

**Summary:**

The paper presents a method to disentangle uncertainties in planning tasks and estimate them separately. It disentangles between epistemic uncertainty---broken into task clarity and task familiarity---and intrinsic uncertainty, defined as the likelihood of successfully producing a plan. The method enables a more accurate and comprehensive uncertainty assessment for LLM-driven planning. Empirically, the paper compares the proposed uncertainty estimation algorithm with SOTA methods used in robot planning and LLMs. Experiments on mobile manipulation and tabletop rearrangement tasks demonstrate accurate and comprehensive uncertainty estimates, with the strongest correlation to actual execution outcomes.

**Questions:**

Q1. In the problem statement, the observation *O* is an "instruction pertaining to the current environment." In my understanding, *O* is a description of the current environment. Is it correct? Is there a realistic example of *O*?

Q2. A follow-up to Q1: Is this method applicable to visual observation? There is an image example in Figure 1. What if you use the image as input to a foundation model with vision capability instead of inputting the text-based environment context?

Q3. How is the success rate measured? If I understand correctly, an "expected success rate" is an estimation of an actual success rate, i.e., task completion rate. Then, how is the actual success rate measured? Do you run the experiments on robots or simulators multiple times and count the number of successful cases?

Q4. How is Equation 4 formulated? Is there a discussion on why the equation in this form is most effective? Are there any ablation studies that show that each component of the equation is effective?

**Ethical Concerns:**

["NO or VERY MINOR ethics concerns only"]

**Final Justification:**

The rebuttal has addressed all my concerns by adding new results and examples. I will increase the score, assuming that they will integrate the results into the paper during revision.

**Limitations:**

L1. Some important terminologies are not clearly defined. For example, there is no formal definition of *epistemic* and *intrinsic* uncertainty. As these terminologies are key components in the paper, a reference to other papers that define the terms may not be sufficient. For clarity, maybe include a few definition blocks in the preliminary section.

L2. While the method estimates uncertainty, it lacks a discussion of how to resolve the uncertainty. For example, using the estimated uncertainty as a reward signal for fine-tuning the LLM or refining the input prompts.

L3. While the empirical results demonstrate a more accurate uncertainty estimation compared with existing baselines, the reliance on neural networks (MLP) leads to a higher computation overhead compared with simple confidence calibration (e.g., KnowNo). This looks like a complexity-accuracy trade-off rather than "outperform" existing baselines.

L4. Some baselines, such as TopK and Vanilla, are too trivial to demonstrate the effectiveness of the proposed method. Instead of showing that the method outperforms the baselines in uncertainty estimation, it is better to demonstrate that the proposed method can improve the task completion rate in some ways.

L5. The appendix does not provide sufficient experimental details. For example, could you please provide real prompts rather than placeholders such as *[What will you do Here]*?

**Paper Formatting Concerns:**

This paper does not have any formatting concerns.

**Quality:**

2

**Strengths And Weaknesses:**

**Strengths**

The paper presents an interesting approach that separately estimates epistemic and intrinsic uncertainty, enabling better uncertainty interpretation in robotic tasks.

Empirical studies demonstrate that the proposed algorithm significantly enhances the accuracy of uncertain estimation, outperforming existing methods in embodied planning and LLM uncertain estimation.

**Weaknesses**

Please see *Limitations* below.

---

> ### Author Rebuttal · Authors · 2025-07-30
>
> Thank you for your thoughtful comments and valuable feedback. Below are our detailed responses.
>
> **Q1. O is a description of the current environment. Is that correct? Are there any real examples of O?**
>
> Yes, you are correct. An example of $O$ is as follows:
>
> > You are a robot operating in an office kitchen. You are in front of a counter with two closed drawers, a top one and a bottom one. There is also a landfill bin, a recycling bin, and a compost bin. On the counter, there is an orange soda, a Pepsi, and an apple.
>
> **Q2. Does this method apply to visual observations? What if the image is used as input for a foundational model with visual capabilities?**
>
> In the specific implementation presented in this paper, which uses LLaMA, images cannot be directly inputted as LLaMA only supports text input. Therefore, it is necessary to first convert the image into a text description using a VLM before feeding it to subsequent models for processing.
>
> However, if a multimodal model supporting images (such as LLaVA) is adopted and the text encoder is replaced with the corresponding multimodal encoder, images can be directly inputted and encoded into feature vectors, keeping the subsequent process unchanged.
>
> **Q3. How is the success rate measured? So, how is the actual success rate measured? Did you conduct multiple experiments?**
>
> The success rate measurement method is as follows:
>
> In the constructed training set and test set, each sample (case) has only one clear binary label: successful cases are marked as 1, and failed cases are marked as 0.
>
> Success is defined as whether the robot's action meets the user's requirements. For example, if the user needs a caffeinated beverage and the robot provides Red Bull, it is considered a success (label 1); otherwise, it is a failure (label 0).
>
> Note that here each sample has only a single binary label, without involving multiple repeated experiments to calculate probabilities.
>
> You understand correctly; the "expected success rate" is indeed an estimate of the actual success rate.
>
> The expected success rate is provided by the model. After training on the training set, the model outputs a value between 0 and 1 as the "expected success rate" for that sample.
>
> Neither the actual success rate nor the expected success rate is statistically calculated through multiple experiments.
>
>
>
> **Q4. How is Equation 4 formulated? Has there been a discussion on why this form of equation is most effective?**
>
> Equation 4 is designed to comprehensively reflect uncertainty in the planning process, combining three main factors: task clarity, expected task success rate, and task similarity.
>
> 1. **Task Clarity**:
> Task clarity is evaluated using two methods: a query-based LLM method and a multi-layer neural network inference method. Both results are represented as $A_{\text{amb}}$, reflecting the ambiguity or difficulty of understanding the task.
>
> 2. **Expected Task Success Rate**:
> Predicted using a multi-layer neural network, yielding success rate $ p $, representing the likelihood of completing the task given the conditions.
>
> 3. **Task Similarity**:
> Estimated using RND to obtain task similarity, $A_{\text{sim}}$, indicating the similarity of the current task to known tasks.
>
> The Equation reflects uncertainty's comprehensive performance under different task conditions through a weighted combination of these three factors.
>
> We formulate Equation 4 as a heuristic model to simulate the generation of uncertainty. It starts from maximum uncertainty (1), subtracting the influence of success rate and task clarity correction to increase certainty, while compensating for extra certainty brought by task familiarity through negative weight parameters. The equation simulates human thinking processes when facing uncertainty, considering the influence of success rate, task clarity, and familiarity comprehensively.
>
> Some ablation study results are integrated into Section 5.1, Table 1. The complete ablation study results are as follows:
>
> |Ablation Condition| Task Clarity | Expected Task Success Rate | Task Similarity | SR-HR-AUC⬆ |
> |-|-|-|-|-|
> |Success Only| | √|| 0.211|
> |Similarity Only||| √ | 0.135 |
> |Clarity only| √ | || 0.173    |
> |Remove Task Similarity| √ | √| | 0.483    |
> |Full Model (All Enabled)| √ | √| √ | 0.534    |
>
>
>
> **L1 & L2. Some important terms are not clearly defined.It lacks discussion on how to resolve uncertainty.**
>
> Thank you for your suggestion. We will add this part of the definitions and discussions to explore possible directions for reducing uncertainty to the paper.
>
> **L3. The reliance on neural networks (MLP) leads to higher computational.**
>
> The computational resources are mainly consumed in the UAN and RND parts, which are both relatively small multi-layer neural networks, resulting in very low computational consumption during actual inference. Specifically, the UAN network and RND network both have a three-layer structure, with neuron numbers of 4096, 128, and 32 respectively. The computational consumption of a single inference is only 0.53 MFLOPs, requiring relatively few computing resources.
>
> Additionally, the computation of feature vectors can utilize the cache after planning by the planning model, avoiding extra computational resource consumption. Although the network training process does require significant resource investment, this part of the computation can be completed in the cloud, not requiring execution on the device, and can be reused after training once.
>
> Overall, although there is an increase in computational resource cost, it is almost negligible. On the other hand, the significant improvement in safety makes this cost increase trivial.
>
> **L4. Some baselines are too trivial to demonstrate the effectiveness of the proposed method.**
>
> Thank you for your suggestion. Since the main focus of this paper is on reducing unexpected actions, we have added some experiments comparing some new metrics, which are defined in IntroPlan[1]. Meanwhile, we tested the IntroPlan method combined with the CURE method.
>
> The experimental results of Overstep Rate, Overask Rate, Help Rate (target success rate 90%) for the IntroPlan method, CURE method, and IntroPlan+CURE method are as follows:
>
> | Metric              | CURE   | Introplan + CURE | Introplan |
> |---------------------|-----------|------------------|-----------|
> | **Overstep Rate** ⬇ | 34.88%    | **21.58%**           | 31.82%    |
> | **Help Rate** ⬇     | 71.33%    | 53.67%           | **19.33%**    |
> | **Overask Rate** ⬇  | **29.44%**    | 53.42%           | 51.72%    |
>
> It can be seen that:
> - In terms of Overstep Rate, the Introplan + CURE method achieved absolutely optimal results, greatly reducing the occurrence of unsafe events.
> - In terms of Help Rate, Introplan is the lowest, indicating it is most confident and requires the least human intervention. However, considering the small gap in Overask Rate between Introplan + CURE and Introplan, it suggests that the help requests of Introplan + CURE are not ineffective, significantly improving safety performance.
> - In terms of Overask Rate, the CURE method is the best, attributed to the initial low success rate of the KnowNo basic method, making the effective help range larger.
>
> **L5. The appendix does not provide enough experimental details.**
>
> You are correct, here is a real prompt:
>
> ### Complete Procedure for Kitchen Operation Experiment
>
> First, we grasp the initial information including the scene and task, for example:
>
> Scene: a bottled unsweetened tea, an orange, and a bag of jalapeno chips
> Task: Put jalapeno chips in the drawer.
>
> Next, we supplement the complete scene information to form a complete scene description:
>
> ```
> We: You are a robot operating in an office kitchen. You are in front of a counter with two closed drawers, a top one and a bottom one. There is also a landfill bin, a recycling bin, and a compost bin.
> We: On the counter, there is an orange soda, a Pepsi, and an apple.
> ```
>
> Based on this, we generate action according to the complete scene and task. This generation process is consistent with KnowNo[2].
> It is emphasized that this generation method is independent of the method proposed in this paper and can adopt any method to generate operation instructions. The generated operation instruction is (if the instruction is not in this format, it will be reformatted by LLM):
>
> ```
> action: pick-up jalapeno chips to bottom drawer
> ```
>
> Next, we combine the aforementioned content into a prompt with specific content:
>
> ```
> You are a human and there is a robot operating in an office kitchen. The robot is in front of a counter with two closed drawers, a top one and a bottom one. There is also a landfill bin, a recycling bin, and a compost bin.
> On the counter, there is {scene}.
> You say: "{task}".
> Then the robot {action}.
> ```
>
> We input this prompt into the Llama3-8B model and extract the last layer's hidden state (last_hidden_state) as the feature vector $T$.
>
> Subsequently, the feature vector is input into the UAN network and RND network to obtain the final expected success rate $p$, ambiguity $A_{\text{amb}}$, and familiarity $A_{\text{sim}}$:
>
> - $p$: 0.962
> - $A_{\text{amb}}$: 0.928
> - $A_{\text{sim}}$: 0.048
>
> Finally, the uncertainty (Uncertainty) is calculated using Eq.4; Confidence is calculated as: 1 - U. The confidence value in the example is 0.378.
>
> **References**
>
> [1] Liang, Kaiqu, Zixu Zhang, and Jaime F. Fisac. "Introspective Planning: Aligning Robots' Uncertainty with Inherent Task Ambiguity." Advances in Neural Information Processing Systems 37 (2024): 71998-72031.
>
> [2] Ren, Allen Z., et al. "Robots that ask for help: Uncertainty alignment for large language model planners." arXiv preprint arXiv:2307.01928 (2023).

---

> > ### Author Response · Authors · 2025-08-05
> >
> > Dear Reviewer:
> >
> > We wanted to express our gratitude for your insightful feedback during the review process of our paper. We hope we have resolved all the concerns and showed the improved quality of our paper. Please do not hesitate to contact us if there are other clarifications we can offer.
> >
> > Best,
> >
> > The authors.

---

> > ### Comment · Reviewer_hD99 · 2025-08-05
> >
> > Thanks for providing the new results and explaining my questions. The rebuttal has addressed the majority of my questions.
> >
> > I have some follow-up questions:
> > 1. According to Eq. 4, the final uncertainty is a combination of multiple uncertainty values, using tunable weights. Is there a reason why the equation is formulated in that structure? E.g., why not use the format ax+by+c?
> >
> > 2. To what degree can this method generalize? Does it work if under the same environment but with a different task? Or the same task with a different environment? For example, tabletop manipulation with different objects.

---

> > > ### Author Response · Authors · 2025-08-06
> > >
> > > **Q1. According to Eq. 4, the final uncertainty is a combination of multiple uncertainty values, using tunable weights. Is there a reason why the equation is formulated in that structure? E.g., why not use the format ax+by+c?**
> > >
> > > The primary reason for adopting the multiplicative structure in Equation 4, $U = 1 - \alpha_1 \cdot (1 - \alpha_2 \cdot A_{\text{amb}}) \cdot p + \alpha_3 \cdot A_{\text{sim}}$, lies in the training process of the Uncertainty Assessment Network (UAN), as described in the paper. Specifically, the UAN is trained differently depending on the clarity of the task objectives:
> > >
> > > - **For tasks with clear objectives**, both task clarity ($A_{\text{amb}}$) and expected success rate ($p$) are trained, allowing the model to learn how these factors jointly influence the uncertainty estimate.
> > > - **For tasks with ambiguous goals**, only task clarity ($A_{\text{amb}}$) is trained, and the expected success rate ($p$) is not included in the training process. This is because ambiguous tasks lack a well-defined success criterion, making it unreliable to estimate the expected success rate.
> > >
> > > The multiplicative form in Equation 4 ensures that the expected success rate ($p$) has a significant impact on the final uncertainty ($U$) only when the task is clear (i.e., when $A_{\text{amb}}$ is low, indicating low ambiguity). In ambiguous scenarios, where $A_{\text{amb}}$ is high, the term $(1 - \alpha_2 \cdot A_{\text{amb}})$ reduces the contribution of $p$, effectively downweighting the influence of an unreliable success rate estimate. This interaction cannot be adequately captured by a linear combination like $ax + by + c$, which would treat all components independently and fail to model the conditional dependence of $p$ on task clarity.
> > >
> > >
> > >
> > > **Q2. To what degree can this method generalize? Does it work if under the same environment but with a different task? Or the same task with a different environment? For example, tabletop manipulation with different objects.**
> > >
> > > Based on our testing, the model demonstrates generalization capability for the same task in different environments. However, when encountering different tasks within the same environment, the model's RND network outputs low familiarity scores, resulting in high uncertainty labels that trigger action rejection.
> > >
> > > It's important to note that CURE, as an uncertainty estimation model, primarily aims to enhance the reliability of embodied AI systems by reducing hazardous and unexpected behaviors. Therefore, when the system encounters tasks outside its training distribution, we recommend rejecting their execution through task familiarity evaluation. While this approach may sacrifice some generalization potential, it significantly improves system reliability by preventing execution of unfamiliar tasks.
> > >
> > > We conducted tests using objects not seen during training, with the following results:
> > >
> > > **Example 1**
> > > Task: "Give me a hamburger"
> > > Action: Pick up the hamburger and hand it to the user
> > > Scene: An orange, an apple, and a hamburger
> > > Success Rate: 0.9565 | Ambiguity: 0.3486 | Task Familiarity: 0.0667 | Confidence: 0.6898
> > >
> > > In this case, despite the hamburger being an unseen object, CURE correctly determined that the instruction was unambiguous, achieving high success (95.65%) with reasonable confidence (0.69).
> > >
> > > **Example 2**
> > > Task: "Give me a hamburger"
> > > Action: Placed the hamburger in the bottom drawer
> > > Scene: An orange, an apple, and a hamburger
> > > Success Rate: 0.3540 | Ambiguity: 0.3025 | Task Familiarity: 0.0796 | Confidence: 0.2102
> > >
> > > This demonstrates a failure case where the same instruction was misinterpreted, resulting in low success (35.4%) and significantly reduced confidence (0.21). These examples collectively show CURE's ability to assess uncertainty and maintain partial generalization with novel objects.
> > >
> > > **Example 3**
> > > Task: "Heat the hamburger"
> > > Action: Placed the hamburger in the microwave
> > > Scene: An orange, an apple, and a hamburger
> > > Success Rate: 0.9311 | Ambiguity: 0.7558 | Task Familiarity: 0.2581 | Confidence: 0.2508
> > >
> > > This scenario involves a completely novel task type ("heat the hamburger") not present in training. The low task familiarity score (0.2581) and confidence (0.2508) reflect the model's appropriate caution, demonstrating its capability to prevent execution of unfamiliar actions despite high success probability, thereby mitigating potential risks.

---

> > > > ### Comment · Reviewer_hD99 · 2025-08-06
> > > >
> > > > Thanks for the response. The rebuttal has addressed my concerns, hence I will increase the score to 4. Please make sure to integrate the new results and examples into the paper during revision.

---

> > > > > ### Author Response · Authors · 2025-08-07
> > > > >
> > > > > Dear Reviewer,
> > > > >
> > > > > Thank you for your thoughtful feedback and for acknowledging that the rebuttal has addressed your concerns. We greatly appreciate your time and effort in reviewing our work. We will ensure that the new results and examples are fully integrated into the revised paper as suggested.
> > > > >
> > > > > Best regards,
> > > > >
> > > > > Authors.

---

### Official Review · Reviewer_mEEn · 2025-06-14

**Clarity:** 2
**Significance:** 3
**Originality:** 3
**Rating:** 4
**Confidence:** 3

**Summary:**

The authors carry out an important and interesting study to better detect LLM uncertainty in robot planning. The authors divide the whole uncertainty into task familiarity and task clarity and then propose methods to estimate the uncertainty of both factors. The task familiarity is estimated by training one distilled NN so that the unexpected instructions naturally cause larger different between distilled NN and the original policy model. Task clarity is assessed by prompting LLM to answer with uni- or multi-goals.

The study and proposed methods are interesting and new. However, there are some questions remained on the soundness of the method, the framework setting, and the testing tasks. Please see the following discussion.

**Questions:**

1) I wonder how the authors construct the training dataset for UAN? How to choose the tasks and plans, also how to judge whether the subtasks are successfully executed?

2) The generalizability may remain a problem. Even for the same set of tasks, expressing with different formats, such as 'picking up sprite' vs. 'I would like to serve the sprite to the user because...'. Whether these different expressions will make the system deviates from the training distribution?

3) As for the two testing environments, it will be better to show specific examples so that the readers can better understand the testing capability of LLMs. For example, is the LLM required to output the full action sequence with pre-defined formats and action options?

4) What will be the reason the robot executes the tasks and fails? Is this due to the wrong action plan or the wrong motion planning? If the motion planning is not grounded to the real world, the testing in the robot scenario is the same as the testing in the digital world like 2D games. Hence, what is the difference of applying this framework to robot with digital games?

**Ethical Concerns:**

["NO or VERY MINOR ethics concerns only"]

**Final Justification:**

I think overall the paper is good in novelty, though the experiments and ablation studies are not enough. I recommend weak accept in this case, while I am also open to other reviewers' decisions and comments.

**Limitations:**

Yes.

**Paper Formatting Concerns:**

No, formatted well.

**Quality:**

3

**Strengths And Weaknesses:**

Strengths: The problem setting is important, to better probe the LLM uncertainty during robot planning to better decrease the risks. The proposed methods to divide the uncertainty and estimate via distilled NN are new and interesting. It is the first time I find studies estimate LLM uncertainty like this work.

Weakness:
1) Why the uncertainty can be divided into task familiarity and task clarity. I guess many uncertainty in task familiarity also relates to task clarity. The two testing tasks in the study is relatively simple. However, there are many hierarchical instructions in robot applications that more complex task and motion planning needed. In these conditions, task familiarity and clarity relates to each other. For example, instead of 'visit building A, then B, then C', 'visit all buildings with shortest path'. It seems the less task clarity will lead to less task familiarity.

2) The way to estimate task familiar and clarity. These methods are new and interesting. However, can there be any clarification to support these designs are reasonable and sound? Why the task should be regarded as unfamiliar if the gaps between original and distilled NNs are larger? Maybe it also results from the ill-trained distilled NN?

3) The testing environments and instructions are simple. It seems no motion planning part is related and the LLM is required only for simple sub-task or even uni-task choices. The generalizability is limited since the trained NN should have pre-designed dataset related to testing tasks. Hence, the online adaptation will not be available.

---

> ### Author Rebuttal · Authors · 2025-07-30
>
> Thank you for your thoughtful comments and valuable feedback. Below are our detailed responses.
>
> **W1. Why can uncertainty be divided into task familiarity and task clarity?**
>
> Thank you for your valuable feedback. The potential correlation between task familiarity and task clarity in complex scenarios, as pointed out, is very reasonable. However, the definition of "task clarity" in this paper differs from the focus you described. The concept of "task clarity" you mentioned is closer to the **level of detail** of the task.
>
> 1. **Core Definition of Task Clarity:** In this paper, task clarity primarily assesses whether the instructions themselves provide sufficient information for the robot to perform the correct operation to fulfill the user's need. Taking the two examples you provided:
>     * "Visit building A, then B, then C."
>     * "Visit all buildings with the shortest paths."
>    Both instructions are sufficiently clear, and the robot can execute the corresponding tasks based on the instructions. However, consider another scenario: the user's need is for the robot to "visit the tallest building," but due to the ambiguity of human language, the operator's actual instruction might be "visit that building" without specifying which one (missing the "tallest" key information). In this case, the robot will struggle to complete the task correctly. The core of task clarity as defined in this paper lies in addressing the consistency between the instruction information and the user's need. Therefore, task clarity does not refer to the level of detail of the instructions, but whether the instruction unambiguously conveys the essential elements needed to complete the task as intended.
>
> 2. **Necessity of Distinguishing Clarity and Familiarity:** Based on the above definition, if the task instructions themselves lack clarity, the robot cannot correctly complete the task, even if it is very familiar with actions like "visiting buildings." Therefore, this paper considers it necessary and meaningful to distinguish task clarity (the clarity of instruction conveying the need) and task familiarity (the proficiency in executing specific types of tasks) as two distinct dimensions.
>
> **W2. Methods for evaluating task familiarity and clarity.  However, is there any explanation to support the rationality and reliability of these designs?**
>
> Your point is very valuable. Regarding the rationality of the task familiarity evaluation method, its design inspiration comes from Random Network Distillation[1]. This method has been widely used in the field of reinforcement learning to assess the novelty of states. The task familiarity metric is mainly used for relative comparison. Indeed, if the distilled neural network is poorly trained, it might lead to an overall high level of gaps corresponding to all tasks, but the relative size differences between different tasks still exist. Therefore, this metric can still effectively reflect the relative familiarity of tasks.
>
> Task clarity can be obtained through supervised learning on the training set or inferred using large language models.
>
> **W3. This method has limited generalization capability because the trained neural network is supposed to contain preset datasets related to the test tasks.**
>
> Your point is very meaningful. The main goal of the paper is to enhance the reliability of embodied intelligent systems and reduce potential dangerous and undesirable behaviors. Therefore, when the system receives tasks outside the training set, opting to refuse execution is a well-considered strategy (our method uses task familiarity evaluation to perform refusal). Although this strategy may limit generalization ability in some situations, we believe this trade-off is worthwhile as it significantly improves the system's reliability and safety.
>
>
>
> **Q1. How the authors constructed the training dataset for UAN. How were tasks and plans selected, and how was the successful execution of sub-tasks judged?**
>
> The construction of the UAN training dataset is mainly achieved through the following two methods:
>
> 1. **Based on LLM Generation:**
>     * Randomly select three objects.
>     * Then, guide the LLM to specify the desired object indirectly. For example, given objects `water`, `orange`, `chips`, the LLM might generate a task description: "I'm thirsty, give me something to quench my thirst."
>     * This constitutes a task sample:
>         * `scene`: Mainly `water`, `orange`, `chips`
>         * `task`: "I'm thirsty, give me something to quench my thirst"
>     * **Success Judgment:** If the agent's action is to give `water` to the user, it is considered successful; otherwise, it is considered a failure.
>
> 2. **Based on Script Generation:**
>     * Directly generate scenes and tasks through scripts. For example, generate a scene containing `water`, `orange`, `chips`, with the task description: "Place the chips next to the fruit."
>     * **Success Judgment:** If the agent's action is to place `chips` next to `orange`, it is considered successful; otherwise, it is considered a failure.
>
> > **Note:** The actual generation process is more complex than the above examples. Please refer to the `dataset_generate.py` file in the code for specific implementation details.
>
> **Regarding the generation of task planning (Plan):**
> The planning generation method adopted in the paper follows the method of the KnowNo[2], which involves querying the LLM to generate task planning.
>
> **Q2. Expressed in different formats, such as "pick up Sprite" vs. "I want to provide Sprite for the user because...". Could these different expressions cause the system to deviate from the training distribution?**
>
> After the LLM completes the planning, we implemented an additional grounding step to align planning actions with robot executable actions. Therefore, the two different expression formats you mentioned will ultimately be unified into the standard form of "pick up Sprite for the user." This step not only ensures the reliability of subsequent execution but also effectively guarantees that the system does not deviate from the training distribution.
>
> For more complex or novel actions that cannot be grounded, considering the safety requirements of the embodied intelligent system, such tasks fall outside the training distribution and will be rejected for execution by the system based on task familiarity assessment. While this approach sacrifices some generalization capability, it significantly enhances the operational safety and reliability of the robot system.
>
> **Q3. Regarding the two test environments, show specific examples to help readers better understand the testing capabilities of the LLM.**
>
> In this experiment, the action format is predefined as either placing an item in a location or giving the item to the user.
>
> Thank you for your suggestion, here are specific examples.
>
> ### Complete Procedure for Kitchen Operation Experiment
>
> Due to length limitations, please refer to the response to item 5 under "Limitations" in "Rebuttal by Authors For Reviewer hD99" for the specific details of this example.
>
> Some brief details are as follows:
>
> Scene: a bottled unsweetened tea, an orange, and a bag of jalapeno chips
> Task: Put jalapeno chips in the drawer.
>
> Based on this, we generate action with KnowNo[2].
>
> Action: pick-up jalapeno chips to bottom drawer
>
> Next, we combine the aforementioned content into a prompt and input this prompt into the Llama3-8B model and extract the feature vector $T$.
>
> Subsequently, the feature vector is input into the UAN and RND network to obtain the final expected success rate $p$, ambiguity $A_{\text{amb}}$, and familiarity $A_{\text{sim}}$:
>
> - $p$: 0.962
> - $A_{\text{amb}}$: 0.928
> - $A_{\text{sim}}$: 0.048
>
> Finally, the confidence value in the example is 0.378.
>
>
> ### Desktop Rearrangement Experiment Process
>
> This process is similar to the kitchen operation experiment, with some specific details as follows:
>
> Scene: blue block, yellow bowl, yellow block, green bowl, green block, blue bowl
> Task: put the yellow block in the yellow bowl.
> Action: put the green block to the back of the blue bowl.
>
> Combined prompt is:
>
> ```
> You are a human and there is a robot. The robot is in front of a table.
> On the table there are these objects: blue block, yellow bowl, yellow block, green bowl, green block, blue bowl.
> The robot is asked to put the yellow block in the yellow bowl.
> Then the robot put the green block to the back of the blue bowl.
> ```
>
> - $p$: 0.181
> - $A_{\text{amb}}$: 0.185
> - $A_{\text{sim}}$: 0.051
>
> Confidence value is 0.109.
>
> **Q4. What are the reasons for the robot failing to execute tasks? What is the difference in applying this framework to robots versus digital games?**
>
> Here is a failed example:
> #### Failure Example
> Scene: a RedBull, a bag of jalapeno chips, and an orange soda
> Task: Bring me something with a kick.
> Action: pick up RedBull
> Success: 0.893
> Ambiguity: 0.056
> Familiarity: 0.084
> Confidence: 0.778
>
> This is a typical failure case: the generated **action** is incorrect, yet the **confidence** is high. Specifically, the task instruction "Bring me something with a kick" is quite ambiguous. The user actually needs "a bag of jalapeno chips," but the model erroneously chose "RedBull," resulting in an excessively high confidence. This is mainly due to the model's inadequate understanding of the instruction and overestimating the instruction clarity.
>
> The uncertainty estimation method proposed in this paper is mainly aimed at the action plan. Therefore, the proposed method is applicable to both robot scenes and digital worlds. And there is no difference because the experiment does not involve motion planning.
>
> **References**
>
> [1] Burda, Y., Edwards, H., Storkey, A., & Klimov, O. (2018). Exploration by random network distillation. arXiv preprint arXiv:1810.12894.
>
> [2] Ren, Allen Z., et al. "Robots that ask for help: Uncertainty alignment for large language model planners." arXiv preprint arXiv:2307.01928 (2023).

---

> > ### Author Response · Authors · 2025-08-05
> >
> > Dear Reviewer:
> >
> > We wanted to express our gratitude for your insightful feedback during the review process of our paper. We hope we have resolved all the concerns and showed the improved quality of our paper. Please do not hesitate to contact us if there are other clarifications we can offer.
> >
> > Best,
> >
> > The authors.

---

### Official Review · Reviewer_FX41 · 2025-06-30

**Clarity:** 3
**Significance:** 2
**Originality:** 2
**Rating:** 4
**Confidence:** 4

**Summary:**

The paper presents CURE, a method for improving the reliability of robot planning using Large Language Models (LLMs) by accurately estimating uncertainty. CURE separates uncertainty into two main categories:
- Epistemic Uncertainty: Split into task familiarity (measured using Random Network Distillation) and task clarity (evaluated via neural networks or direct LLM queries).
- Intrinsic Uncertainty: Based on the expected success rate influenced by environmental factors.

The authors demonstrate the effectiveness of CURE in two robotic scenarios (kitchen manipulation and tabletop rearrangement), showing that it significantly outperforms baseline methods by providing uncertainty estimates better aligned with actual task outcomes.

**Questions:**

- Could you explicitly report the original metrics (such as overall success rates without uncertainty-based intervention) for IntroPlan and KnowNo?
- Could you provide additional analysis explaining how your proposed metrics (SR-HR-AUC and Spearman correlation) interact differently with existing baselines and discuss specific scenarios or limitations where these metrics might misrepresent actual baseline performance?
- Given that IntroPlan already implicitly addresses epistemic uncertainty via introspective knowledge-base retrieval, could you clarify the practical advantages and potential trade-offs (e.g., scalability, complexity) of explicitly modeling task similarity, ambiguity, and expected success rate using separate neural networks? Additionally, can you provide qualitative examples illustrating success and failure cases of your method (CURE)?

**Ethical Concerns:**

["NO or VERY MINOR ethics concerns only"]

**Final Justification:**

The author addressed most of the concerns during the rebuttal. I think the paper will be strengthened if the author incorporates all the additional results and discussions into the paper.

**Limitations:**

Yes. The authors discussed limitations.

**Paper Formatting Concerns:**

No major formatting issues

**Quality:**

2

**Strengths And Weaknesses:**

Strengths:
- The author proposes an interesting method that differentiates between epistemic and intrinsic uncertainties.
- Proposes SR-HR-AUC and Spearman correlation to quantify uncertainty estimation performance
- Demonstrates improvement over existing baselines in two diverse scenarios: kitchen manipulation and tabletop rearrangement tasks.

Weakness:
1. While the paper proposes novel metrics (SR-HR-AUC and Spearman correlation) to assess uncertainty estimation, it's crucial to also present and analyze previously established metrics (like overall success rate) clearly. Omitting these original metrics can lead to incomplete evaluations or potentially biased comparisons.
Specifically, regarding the IntroPlan baseline, the authors briefly attribute its low performance on the new metrics to a high initial success rate, yet they don't explicitly report or quantify this original success rate. Without this information, it is challenging to fairly assess the baseline performance.
To address this, the authors should explicitly:
- Report original metrics in IntroPlan and KnowNo (e.g. success rate)
- Provide more analysis why these new metrics negatively influence some baseline performance.
- Discuss limitations or scenarios where the proposed metrics might misrepresent certain baselines.

Including these analyses would strengthen the evaluation.

2. It’s worth including some qualitative examples to better understand how the proposed method is effective and the failure cases of the proposed method.

3. I highly doubt the necessity of training two additional networks to explicitly predict task similarity, task ambiguity, and expected success rate. Introspective planning already effectively leverages prior knowledge from the knowledge base, implicitly reducing epistemic uncertainty by retrieving relevant historical interactions that inherently capture task similarity and clarity. Although explicitly modeling these factors through additional neural networks appears interesting, it might be unnecessary and potentially less scalable compared to IntroPlan. Therefore, I suggest the authors add a more thorough discussion about these trade-offs, provide a fairer empirical comparison with prior works (especially IntroPlan) and clearly articulate which specific aspects or components of CURE offer practical advantages.

---

> ### Author Rebuttal · Authors · 2025-07-30
>
> Thank you for your thoughtful comments and valuable feedback. Below are our detailed responses.
>
> **W1. Provide further analysis explaining why these new metrics negatively impact some baseline performances.**
>
>
> Regarding the poor performance of IntroPlan on the new metrics, our hypothesis is that it possesses more comprehensive information, making the model overly confident in its chosen answers. Below are the confidence distribution statistics for IntroPlan and KnowNo in the test questions:
>
> **KnowNo:**
> - Average Confidence: 0.320416
> - Minimum Confidence: 0.214667
> - Maximum Confidence: 0.498671
> - Standard Deviation: 0.054888
>
> **IntroPlan:**
> - Average Confidence: 0.905210
> - Minimum Confidence: 0.300099
> - Maximum Confidence: 1.000000
> - Standard Deviation: 0.198491
>
> In IntroPlan, there are 242 items with confidence values exceeding 0.99, which leads to low differentiation in confidence levels. Our new metrics (SR-HR-AUC and Spearman correlation) mainly focus on the differentiation of confidence between successful and failed tasks, thus resulting in poor performance for IntroPlan on these two metrics.
>
>
>
> **Q1. Can you explicitly report the original metrics for IntroPlan and KnowNo?**
>
> Figure 6 in the original text shows the overall success rate when there is no uncertainty-based intervention. When the Help Rate is 0, it represents the overall success rate. The specific success rates are as follows:
>
>
> |  Method        | KnowNo | CURE  | IntroPlan |
> |--------------------|--------|-------|-----------|
> | Initial Success Rate ↑ | 40.67% | 40.67% | 63.00%    |
>
> Since CURE is a subsequent step based on the planning results of KnowNo, the original metrics should be consistent with KnowNo. Therefore, no additional explanation is provided in the paper.
>
> **Q2. Can you provide additional analysis explaining how your proposed metrics interact differently with existing baselines and discuss specific scenarios or limitations where these metrics might distort actual baseline performance?**
>
> The metrics we propose differ significantly from the objectives of existing baseline tests. Existing baselines primarily focus on the overall success rate of tasks, aiming to enhance the performance of planning models. In contrast, our proposed metrics (SR-HR-AUC and Spearman correlation) emphasize the accuracy of uncertainty to prevent robots from exhibiting unexpected behavior. This metric is similar to the Overstep Rate in IntroPlan but still differs:
>
> Overstep Rate is defined as the proportion of overconfident or incorrect options generated by the planner when certain. It is calculated as: Count (robot is certain but wrong)/Count (robot is certain).
>
> We conducted additional experiments to evaluate the Overstep Rate, Overask Rate, and Help Rate of the IntroPlan method, CURE method, and IntroPlan+CURE method when the target success rate is 90%. The experimental results are shown below:
>
> | Metric            | CURE   | IntroPlan + CURE | IntroPlan |
> |-------------------|--------|------------------|-----------|
> | **Overstep Rate** ⬇  | 34.88% | **21.58%**      | 31.82%    |
> | **Help Rate** ⬇     | 71.33% | 53.67%          | **19.33%** |
> | **Overask Rate** ⬇   | **29.44%** | 53.42%       | 51.72%    |
>
> The results show that the IntroPlan + CURE method achieved the absolute optimal effect in terms of Overstep Rate, significantly reducing unsafe incidents. In terms of Help Rate, IntroPlan demonstrated the lowest rate, indicating higher confidence and reduced need for human intervention. However, considering the small difference between IntroPlan + CURE and IntroPlan in the Overask Rate metric, it suggests that the help behavior of IntroPlan + CURE was not ineffective, significantly improving safety performance. CURE performed best in the Overask Rate, partly due to the lower initial success rate of its base method, KnowNo, making the effectiveness of the help more significant.
>
> **Q3. Given that IntroPlan implicitly addresses epistemic uncertainty through introspective knowledge base retrieval, can you clarify the actual advantages and potential trade-offs (e.g., scalability, complexity) of explicitly modeling task similarity, ambiguity, and expected success rate using separate neural networks? Additionally, can you provide qualitative examples to illustrate the success and failure cases of your method (CURE)?**
>
> Compared to IntroPlan, the proposed method has significant advantages in scalability. A typical example is that IntroPlan can be integrated with CURE to further enhance system performance. As demonstrated in the experiments concerning the previous question, when IntroPlan is used in conjunction with CURE, both the sr_hr_area and Overstep Rate (with a target success rate of 90%) improve compared to using IntroPlan alone. This indicates that there is still room for improvement in the accuracy of uncertainty estimation in IntroPlan, and by introducing CURE, this accuracy can be further optimized.
>
> Regarding complexity, the main disadvantage of the method proposed in this paper is the need for a large training set for model training. However, considering that a mature robotic system typically requires numerous training cases for pre-training, this limitation has relatively little impact on practical applications, making this disadvantage acceptable in practice.
>
> Examples of CURE are as follows:
>
> #### Success Example 1
> - **Scene**: a bottled unsweetened tea, an orange, and a bag of jalapeno chips
> - **Task**: Put jalapeno chips in the drawer.
> - **Action**: pick-up jalapeno chips to bottom drawer
> - **Success**: 0.9624221324920654
> - **Ambiguity**: 0.9281229376792908
> - **Familiarity**: 0.04835970117710531
> - **Confidence**: 0.3781147508416325
>
> #### Success Example 2
> - **Scene**: a bag of kettle chips, an apple, and a Pepsi
> - **Task**: Put that apple in the bottom drawer.
> - **Action**: move apple to bottom drawer
> - **Success**: 0.9762726426124573
> - **Ambiguity**: 0.17079491913318634
> - **Familiarity**: 0.03660911461338401
> - **Confidence**: 0.8396180854178965
>
> In the above two examples, the first example has a high task ambiguity due to the lack of specification on which drawer. Despite high expected success and familiarity, the execution confidence remains low. The second example clearly specifies "bottom drawer," significantly reducing task ambiguity. Although the expected success and task familiarity are similar to the first example, the execution confidence is significantly improved due to the clearer task instructions.
>
> #### Failure Example
>
> - **Action**: pick up RedBull
> - **Scene**: a RedBull, a bag of jalapeno chips, and an orange soda
> - **Task**: Bring me something with a kick.
> - **Success**: 0.8931169509887695
> - **Ambiguity**: 0.056598808616399765
> - **Familiarity**: 0.08403261657804251
> - **Confidence**: 0.7787547493353486
>
> This is a typical failure example where the planned action is incorrect, but the confidence is high. The instruction is quite ambiguous here; the user actually wanted a bag of jalapeno chips, but the model mistakenly interpreted it as requiring RedBull. This reflects a lack of understanding from the model, failing to correctly interpret the instruction while erroneously perceiving it as clear.

---

> > ### Author Response · Authors · 2025-08-05
> >
> > Dear Reviewer:
> >
> > We wanted to express our gratitude for your insightful feedback during the review process of our paper. We hope we have resolved all the concerns and showed the improved quality of our paper. Please do not hesitate to contact us if there are other clarifications we can offer.
> >
> > Best,
> >
> > The authors.

---

> > ### Comment · Reviewer_FX41 · 2025-08-06
> >
> > Thanks to the authors for their detailed response and the additional experimental results, which are very helpful. However, I still have concerns regarding the trade-off between success rate and help rate for CUBE, KnowNo, and IntroPlan. Figure 6 indicates that IntroPlan dominates this trade-off, which demonstrates significantly better performance compared to the other methods, particularly when the help rate is low. To better show these differences in performance, I suggest consolidating the success rate vs. help rate curves for all baselines and CUBE into a single, unified plot rather than displaying them separately. This would enhance the clarity and interpretability of the comparative analysis.
> >
> > As the authors cannot update figure results during rebuttal, they should at least describe the resulting figure, clarify whether IntroPlan dominates, and clearly articulate how and why the proposed CUBE approach is superior or offers distinct advantages.

---

> > > ### Author Response · Authors · 2025-08-07
> > >
> > > Thank you for your valuable feedback. Regarding your concerns about the **trade-off between Success Rate and Help Rate for CURE, KnowNo, and IntroPlan**, we understand your points and respond as follows:
> > >
> > > 1. **On Figure 6 and IntroPlan’s Advantages:**
> > >    You correctly noted that **Figure 6 shows IntroPlan’s superior performance on the Success Rate vs. Help Rate trade-off curve, particularly in low Help Rate regions**. This indeed reflects IntroPlan’s strong foundational planning capabilities, with an initial Success Rate (63%) significantly higher than KnowNo-based CURE (40.67%). In scenarios requiring no or minimal human assistance (i.e., low Help Rate), IntroPlan maintains a high Success Rate, which is a direct advantage of its design.
> > >
> > > 2. **Improved Visualization:**
> > >    We fully agree with your suggestion. In the revised version of the paper, **we will consolidate the Success Rate vs. Help Rate curves for CURE (KnowNo-based), KnowNo, IntroPlan, and the newly proposed IntroPlan + CURE method into a single unified figure**. This will allow readers to clearly see the trade-off characteristics of each method and IntroPlan’s advantage in low Help Rate regions.
> > >
> > >    The analysis of the figure is as follows:
> > >    1. **IntroPlan’s Initial Advantage**: At a Help Rate of 0%, IntroPlan and IntroPlan + CURE achieve an initial Success Rate of 63.00%, significantly higher than KnowNo and CURE(KnowNo-based）’s 40.67%. This indicates IntroPlan’s strong task completion capability without human intervention.
> > >    2. **Safety Improvement with IntroPlan + CURE**: As the Help Rate increases, IntroPlan + CURE shows a significant improvement in Success Rate compared to IntroPlan alone, demonstrating that CURE’s more accurate uncertainty estimation markedly enhances Success Rate under human intervention.
> > >    3. **Comparison with KnowNo and CURE (KnowNo-Based)**: Although KnowNo and CURE (KnowNo-based) also achieve a relatively high SR_HR_AUC, their lower initial Success Rate compared to IntroPlan makes them less competitive.
> > >
> > > 3. **CURE’s Value and Unique Advantages:**
> > >    While IntroPlan excels in baseline Success Rate, our CURE method and its integration with IntroPlan provide unique and significant value, primarily in **improved uncertainty estimation accuracy and enhanced safety**.
> > >    * **Lowest Overstep Rate (21.58% vs. IntroPlan 31.82%)**: This indicates a significantly reduced probability of errors when the model is confident in its plan, **greatly enhancing system safety** and avoiding failures due to overconfidence. This is a direct benefit of CURE’s uncertainty estimation capabilities.
> > >    * **Higher SR_HR_AUC (0.525 vs. IntroPlan 0.005)**: This demonstrates a **qualitative leap in the calibration of uncertainty estimation** for IntroPlan + CURE. Its confidence scores more reliably reflect actual success probabilities, making strategies based on confidence-driven help requests more effective.
> > >    * **Maintaining High Baseline Success Rate**: IntroPlan + CURE inherits IntroPlan’s high initial Success Rate (same as IntroPlan).
> > >
> > > To prevent ambiguity, in subsequent versions of the paper, we will switch CURE to use IntroPlan's planning results for its estimations. We believe this will perfectly address your concern.
> > >
> > > It is noteworthy that CURE and IntroPlan are not entirely opposing methods. **CURE can build upon IntroPlan to further enhance the robotic system's awareness of its planning reliability.** Crucially, it achieves this **without sacrificing the baseline success rate**, while simultaneously substantially improving safety (Overstep Rate ↓) and uncertainty calibration (SR_HR_AUC ↑↑). **This represents the unique value and core advantage of the CURE method.**

---

> > > > ### Author Response · Authors · 2025-08-08
> > > >
> > > > We sincerely appreciate your time and expertise throughout the review process. Your constructive feedback has provided valuable perspectives for refining our work. We stand ready to provide further clarification should any questions arise during the final assessment.
> > > > ﻿
> > > >
> > > > Best regards,
> > > >
> > > > The Authors

---

### Official Review · Reviewer_9hBH · 2025-07-03

**Clarity:** 2
**Significance:** 3
**Originality:** 3
**Rating:** 3
**Confidence:** 4

**Summary:**

The paper introduces CURE, a framework for estimating uncertainty of LLM-based planning more accurately. It separates uncertainty into task clarity, familiarity, and environmental risk, leveraging random network distillation to empirically estimate uncertainties. Tested on robot tasks, CURE outperforms existing methods by better predicting when plans may fail.

**Questions:**

How sensitive is CURE’s performance to the quality or size of the prior task dataset used for RND?
What happens if task familiarity data is unavailable—can CURE still function effectively without RND?
Have you tested generalization to new task types or domains not seen during training? Can you provide some examples of the tasks?
Could you share more about the labeling effort required for training UAN (e.g., how many success/clarity labels were needed)?
How would you calibrate the predictions?
Why use different models (70B vs 8B) for different tasks? How would one pick given a new domain?

**Ethical Concerns:**

["NO or VERY MINOR ethics concerns only"]

**Final Justification:**

The conceptual novelty of the proposed method is incremental. The method relies on multiple heuristic and data-dependent modules that make the system bulky and complex to train/tune and deploy. These overheads undermine the claimed plug-and-play nature of the approach, raising concerns about scalability and real-world practicality. The performance gain over KnowNo seems marginal in certain domains, while it removes the formal guarantees conformal prediction (KnowNo) provides. I won't strongly oppose the acceptance of this work if all other reviewers and the AC finds value in this work, hence would like to maintain weak reject.

**Limitations:**

yes

**Quality:**

2

**Strengths And Weaknesses:**

Strengths:
- Fine-grained decomposition of uncertainty into task clarity, familiarity, and intrinsic uncertainty
- Modular, plug-and-play design requiring no changes to the LLM planner
- Compared with a range of baseline methods and show strong empirical performance across two task domains

Weaknesses:
- CURE requires more input data than baselines, including task clarity labels, execution outcomes, and prior tasks; especially prior tasks may not be easy to obtain given that LLMs are trained on internet-scale data
- CURE involves supervised training of multiple components and tuning many hyperparameters; demands more computational resources and engineering effort than lightweight, prompt-based baselines
- CURE does not generalize to new or unseen task domains without retraining

---

> ### Author Rebuttal · Authors · 2025-07-30
>
> Thank you for your thoughtful comments and valuable feedback. Below are our detailed responses.
>
> **W1. Requires more input data compared to baseline methods. acquiring prerequisite tasks might be challenging.**
>
> We can obtain task datasets by simulating tasks in virtual environments, which is a common and effective way to gather data. In this paper, we employed the following two dataset generation methods:
>
> 1. LLM-based generation: We first randomly select three items and then use a large language model to indirectly indicate the desired option. For example, if the selected items are water, oranges, and chips, the LLM might say, "I am thirsty; give me something to quench my thirst." This constitutes a task dataset.
>
> 2. Script-based direct generation: For instance, we generate water, oranges, and chips and set the task as placing the chips next to the fruits.
>
> For more complex tasks, we can use virtual simulation environments to generate data and combine them with real-world datasets. While obtaining data for prerequisite tasks may present certain challenges, we believe these methods can effectively address the issue.
>
> Moreover, in a mature robotic system, pre-training typically requires numerous training examples. These pre-training examples can serve as prior tasks, thereby resolving the issue in practical applications.
>
> **W2. Training and Tuning of hyperparameters, resulting in higher computational and engineering costs.**
>
> The primary computational cost comes from the UAN and RND components, both of which are relatively small multi-layer neural networks. Consequently, the inference cost is very low. Specifically, the UAN and RND networks are three-layer neural networks with 4096, 128, and 32 neurons, respectively. Each network inference consumes only 0.53 MFLOPs.
>
> The feature vector computation can leverage caching at the end of the planning model’s execution, so it does not incur additional computational overhead. While training the networks indeed requires considerable resources, this process can be conducted in the cloud, eliminating the need for on-device computation. Moreover, the trained model can be reused multiple times. Overall, although the computational resource cost increases, it is justified given the improvements in safety.
>
> Regarding hyperparameter optimization, this paper uses the same hyperparameter settings for two distinct tasks, and the experimental results demonstrate that this configuration achieves good performance. Therefore, there is no need to spend too much effort on hyperparameter tuning.
>
> **W3. Cannot generalize without retraining.**
>
> This is indeed an important consideration. However, the main goal of this paper is to enhance the reliability of embodied intelligence systems, reducing dangerous and unintended behaviors. Therefore, when the system encounters tasks outside the training set, we recommend refusing to execute them. This can be achieved by evaluating task familiarity. While this approach may sacrifice generalization capability, by refusing to perform unfamiliar tasks, we can significantly enhance the system's reliability.
>
>
>
> **Q1. How sensitive is the performance of the CURE model to the quality or scale of prior task datasets used for RND?**
>
> We conducted experiments on dataset scale, and the results are as follows:
>
> |Training Set Size|100|300|1000|3000|10000|30000|100000|
> |---|---|---|---|---|---|---|---|
> |SR-HR-AUC|0.125|0.146|0.174|0.234|0.362|0.430|0.547|
>
> The experimental results demonstrate that increasing the dataset scale significantly improves the performance of the CURE model. When the dataset reaches about 10,000 samples, the performance improvement becomes noticeable, and further scaling up to 100,000 samples still yields continuous gains.
>
> Additionally, we investigated the impact of dataset quality on the model. The datasets in this paper were generated both via LLM (large language model) automation and manual definition. For specific generation methods, please refer to our response in W1. The model performed well on datasets constructed using these methods, indicating that CURE’s performance is relatively less dependent on data quality. In other words, the model can effectively utilize automatically generated datasets to achieve robust performance.
>
> **Q2. Can CURE still function effectively without RND, i.e., in the absence of task familiarity data?**
>
> Yes. In Table 1, we present the experimental results of the CURE algorithm without using RND (i.e., CURE w/o sim). These results show that even without task familiarity data, CURE outperforms baseline methods and maintains a certain level of advantage.
>
> **Q3. Have you tested generalization to new task types or domains unseen during training?**
>
> We tested on objects that were unseen during the training phase, and the results are as follows:
>
> 1. Task: Give me a hamburger
>    Action: Pick up the hamburger and give it to the user.
>    Scene: An orange, an apple, and a hamburger.
>    Success Rate: 0.9565
>    Ambiguity: 0.3486
>    Task Familiarity: 0.0667
>    Confidence: 0.6898
>
>    In this example, although the hamburger is an unseen object, CURE can determine that the task instruction "give me a hamburger" is relatively clear, achieving a high success rate with a confidence level of 0.69.
>
> 2. Task: Give me a hamburger
>    Action: Pick up the hamburger and place it in the bottom drawer.
>    Scene: An orange, an apple, and a hamburger.
>    Success Rate: 0.3540
>    Ambiguity: 0.3025
>    Task Familiarity: 0.0796
>    Confidence: 0.2102
>
>    In this example, the same task instruction "give me a hamburger" was executed incorrectly, with the model mistakenly placing the hamburger in the bottom drawer. The success rate is low (0.3540), and confidence drops to 0.21. These two examples demonstrate that the CURE model can accurately assess uncertainty even when handling unseen objects, showcasing a degree of generalization capability.
>
> 3. Task: Heat the hamburger
>    Action: Pick up the hamburger and place it in the microwave.
>    Scene: An orange, an apple, and a hamburger.
>    Success Rate: 0.9311
>    Ambiguity: 0.7558
>    Task Familiarity: 0.2581
>    Confidence: 0.2508
>
>    The third example illustrates another scenario where the task instruction "heat the hamburger" is unseen in the training set, which is a new task type. Since this task is unfamiliar, the task familiarity is very low, leading to low confidence. This result reflects the model's ability to prevent execution of unknown actions, thereby avoiding potential risks.
>
> **Q4. Can you provide some example tasks? How much annotation effort (e.g., success/clarity labels) is required to train UAN?**
>
> In this paper, we generated 100,000 training samples.
>
> Crucially, the generation of success/clarity labels for these 100,000 samples was achieved through automated methods (LLM simulation, scripted checks in simulation), requiring minimal manual intervention. This significantly reduces the practical labeling burden.
>
> Additionally, we tested the results with varying training dataset sizes; please refer to the response to Q1 for detailed results.
>
> Here are some example tasks:
>
> #### Success Example 1
>
> - Scene: A bottled unsweetened tea, an orange, and a bag of jalapeño chips.
> - Task: Put jalapeño chips in the drawer.
> - Action: Pick up jalapeño chips and place them in the bottom drawer.
> - Success: 0.9624
> - Ambiguity: 0.9281
> - Familiarity: 0.0484
> - Confidence: 0.3781
>
> #### Success Example 2
>
> - Scene: A bag of kettle chips, an apple, and a Pepsi.
> - Task: Put the apple in the bottom drawer.
> - Action: Move the apple to the bottom drawer.
> - Success: 0.9763
> - Ambiguity: 0.1708
> - Familiarity: 0.0366
> - Confidence: 0.8396
>
> In the first example, the task does not explicitly specify which drawer, resulting in higher task ambiguity. Despite relatively high success and familiarity rates, the execution confidence remains low. In the second example, the task explicitly specifies "bottom drawer," significantly reducing task ambiguity. Although the expected success rate and task familiarity are similar to the first example, the clearer task instruction leads to significantly higher execution confidence.
>
> #### Failure Example
>
> - Scene: A RedBull, a bag of jalapeño chips, and an orange soda.
> - Task: Bring me something with a kick.
> - Action: Pick up RedBull.
> - Success: 0.8931
> - Ambiguity: 0.0566
> - Familiarity: 0.0840
> - Confidence: 0.7788
>
> This is a typical failure case where the planned action is incorrect, yet the confidence is high. Here, the instruction is somewhat ambiguous; the user actually wants a bag of jalapeño chips, but the model incorrectly interprets the instruction and selects RedBull. This reflects the model’s limited understanding and failure to correctly interpret the instruction while mistakenly assuming the instruction is clear.
>
> **Q5. How to calibrate?**
>
> Our approach involved splitting the test set into two parts: one for calibration and the other for testing. Given a target success rate, we first determined a threshold on the calibration set. Tasks with confidence below the threshold were completed with human assistance, while tasks with confidence above the threshold were executed autonomously. By adjusting the threshold, we could make the success rate approach the target success rate, thereby achieving calibration.
>
> We tested this method for a target success rate of 85%. The results showed that the success rate on the test set reached 84.33%, differing from the target by only 0.67%. This demonstrates that the CURE method achieves good calibration accuracy.
>
> **Q6. Why were different models used for different tasks?**
>
> Since the second tabletop placement task was relatively simple, we achieved satisfactory results with the 8B model. To save computational resources, we opted to use the 8B model.
>
> **Q7. How should the model be selected when encountering new domains?**
>
> If computational resources allow, selecting a larger model generally leads to better performance.

---

> > ### Author Response · Authors · 2025-08-05
> >
> > Dear Reviewer:
> >
> > We wanted to express our gratitude for your insightful feedback during the review process of our paper. We hope we have resolved all the concerns and showed the improved quality of our paper. Please do not hesitate to contact us if there are other clarifications we can offer.
> >
> > Best,
> >
> > The authors.

---

> > ### Comment · Reviewer_9hBH · 2025-08-06
> > **Thanks for the response**
> >
> > Thank the authors for the responses to my questions.
> >
> > - I am not convinced that it is easy to generate the dataset required to train the proposed model. To obtain end-to-end task success rates, you need to perform physical rollouts to determine that, no? At the same time, success rates cannot be determined without taking the scene geometry into account e.g. if an item is visible but not within the robot's reach, how do you take that into account?
> > - The provided example "Bring me something with a kick." is very confusing, it seems that "redbull" would be a plausible option, why count that as failure? how did you know "the user actually wants a bag of jalapeño chips"? It seems wrong to pick an answer first, then add ambiguity to generate the ambiguous version of the task and expect the model to pick the preselected answer.

---

> > > ### Author Response · Authors · 2025-08-07
> > >
> > > Thank you for your continued attention to our paper and for providing valuable feedback. We address your latest comments below:
> > >
> > > **On the Difficulty of Generating the Dataset**
> > >
> > > In current embodied AI research, tasks are typically divided into task planning and motion execution. Our work (and related prior works, such as KnowNo [1] and IntroPlan [2]) focuses primarily on uncertainty estimation during the task planning phase. Thus, task success is determined when the robot’s planned task aligns with the user’s intended goal, rather than requiring physical simulations. This approach is common in task planning studies, as it effectively evaluates planning performance while avoiding the high cost and complexity of physical experiments.
> > >
> > > Regarding your point about scene geometry (e.g., an item being visible but out of the robot’s reach), this is a highly valuable observation that highlights a gap between uncertainty estimation and real-world application. Our CURE framework currently focuses on uncertainty in task planning, but we acknowledge that uncertainties in motion execution (e.g., reachability or environmental constraints) are equally critical in embodied AI systems. We plan to extend the CURE framework in future work to incorporate uncertainty estimation for motion execution, creating a more comprehensive uncertainty estimation pipeline.
> > >
> > > **On the “Bring me something with a kick” Example**
> > >
> > > **1. Logic Behind the Example and “Failure” Determination**
> > >
> > > You correctly noted that “RedBull” is a reasonable response to “something with a kick”. This observation precisely underscores the core challenge we aimed to address: the impact of linguistic ambiguity on robotic task execution. Below, we outline the logic behind this example:
> > >
> > > - **Task Design and Intent:** To simulate the impact of linguistic ambiguity, we defined a scenario where a user in a kitchen wants a spicy snack (e.g., jalapeño chips) but uses the ambiguous phrase “something with a kick.” This phrase could mean “spicy” (aligned with the intent) or “energizing” (e.g., RedBull).
> > > - **Logic for “Failure” Determination:** While selecting “RedBull” is linguistically reasonable, it deviates from the predefined physical target (jalapeño chips). Thus, “RedBull” is deemed a “failure”.
> > > - **Clarification:** Our goal is **not to require the model to guess “chips”** but to evaluate its ability to **perceive uncertainty** in its decisions. Selecting “RedBull” highlights the risk of intent misalignment due to ambiguity, which is exactly what the experiment is designed to test.
> > >
> > > We believe that in real-world embodied AI applications, **safety and reliability take precedence over blind execution**. CURE’s uncertainty quantification enables the system to opt for safer strategies (e.g., seeking clarification or refusing to act) when faced with ambiguous instructions.
> > > Thus, for an ambiguous instruction like “Bring me something with a kick,” the ideal model behavior would be:
> > >   - Detecting high uncertainty (due to the instruction’s ambiguity in the given context) and outputting a **significantly lower confidence score**.
> > >   - When confidence falls below a safety threshold, **proactively refusing to act blindly** and instead requesting clarification (e.g., “Do you mean something spicy or something energizing?”).
> > >
> > > **2. Response to Concerns About “Preselecting an Answer and Adding Ambiguity”**
> > >
> > > This design is not intended to force the model to match a preselected answer but to **study the risk of intent misalignment caused by linguistic ambiguity**.
> > > Specifically:
> > > - **Why Predefine an Intent?** In real interactions, users may have a clear intent but express it ambiguously. We simulate this by predefining a scenario intent (chips) to test the model’s response to ambiguous instructions.
> > > - **Why Introduce Ambiguity?** The ambiguous instructions reflects the characteristics of human language. This design enables us to quantify the model’s performance under ambiguity and assess whether it can identify task uncertainty.
> > > - **Experimental Goal:** The experiment assesses CURE’s robustness and safety under ambiguity, not its ability to guess a “correct” answer.
> > >
> > >
> > > Sincerely,
> > >
> > > Authors.
> > >
> > >
> > > [1] Ren, A. Z., Dixit, A., Bodrova, A., Singh, S., Tu, S., Brown, N., ... & Majumdar, A. (2023). Robots that ask for help: Uncertainty alignment for large language model planners. arXiv preprint arXiv:2307.01928.
> > >
> > > [2] Liang, K., Zhang, Z., & Fisac, J. F. (2024). Introspective Planning: Aligning Robots’ Uncertainty with Inherent Task Ambiguity. Advances in Neural Information Processing Systems, 37, 71998-72031.

---

> > > > ### Author Response · Authors · 2025-08-08
> > > >
> > > > We sincerely appreciate your time and expertise throughout the review process. Your constructive feedback has provided valuable perspectives for refining our work. We stand ready to provide further clarification should any questions arise during the final assessment.
> > > > ﻿
> > > >
> > > > Best regards,
> > > >
> > > > The Authors

---

### Author Response · Authors · 2025-08-06
**General Response**

**General Response**

Dear Reviewers and Area Chair,

Thank you for your valuable feedback on our manuscript, "Towards Reliable LLM-based Robot Planning via Combined Uncertainty Estimation (CURE)." CURE enhances LLM-based robot planning by decomposing uncertainty into epistemic (task clarity, familiarity) and intrinsic components, improving alignment with execution outcomes. We appreciate your recognition of its novel uncertainty decomposition (9hBH, FX41, mEEn, hD99), strong performance (9hBH, hD99), and significance (mEEn, FX41). Based on your feedback we’ve revised the manuscript as follows:

**New Experiments:**
- **CURE + IntroPlan Integration**: Combined CURE with IntroPlan, achieving a lower Overstep Rate (21.58%) compared to IntroPlan (31.82%) and CURE alone (34.88%) at 90% success rate (Rebuttals to 9hBH, FX41, hD99).
- **RND Scale Sensitivity**: Performance improves with prior task dataset size (100 to 100,000 samples), with SR-HR-AUC rising from 0.125 to 0.547 (Rebuttal to 9hBH, Q1).
- **Generalization to Unseen Tasks/Objects**: CURE accurately handles novel objects (e.g., hamburger) and tasks (e.g., "heat the hamburger"), safety by rejecting unfamiliar tasks (Rebuttals to 9hBH, mEEn, hD99).
- **Ablation Studies on Equation 4**: Full model achieves highest SR-HR-AUC (0.534) compared to partial configurations (Rebuttal to hD99, Q4).
- **Calibration Accuracy**: At 85% target success rate, CURE achieves 84.33% on test set (Rebuttal to 9hBH, Q5).

**Clar:**
- **Terminology**: Added definitions for epistemic and intrinsic with references and context (Rebuttal to hD99, L1).
- **Equation 4 Rationale**: Explained multiplicative form (U = 1 - S * (1 - C) * (1 - w * F)) for better dependency modeling (Rebuttal to hD99, follow-up Q1).
- **Task Clarity vs. Familiarity**: Clarified distinction address separate uncertainty sources (Retal to mEEn, W1).
- **Data Generation**: Detailed automated data generation (100,000 samples) minimal manual labeling (Rebuttals to mEEn Q1, 9hBH Q4).
- **CURE vs. IntroPlan**: Discussed explicit vs. implicit uncertainty modeling, highlighting CURE’s scalability and differentiation (Rebuttal to FX41, Q3).

**Additional Notes:**
The revised manuscript addresses concerns on generalization (mEEn, hD99), metrics (FX41), and complexity (9hBH, hD99). We believe these updates strengthen CURE’s contribution to the NeurIPS community and kindly request the Area Chair’s consideration.

Thank you,

Authors

---

### Decision · Program_Chairs · 2025-09-17

**Decision:**

Accept (poster)

**Comment:**

This paper proposes a finer-grained uncertainty quantification analysis in LLM planning inspired by RL: epistemic uncertainty and intrinsic uncertainty. The former concerns uncertainty at the task level, whereas the latter is about uncertainty around executing a fixed plan. The paper proposes a system called CURE for explicitly modeling these two types of uncertainty, which they evaluate in kitchen manipulation and tabletop settings.

This paper is quite borderline, but the authors did a thorough job during the discussion period to address a majority of the reviewer concerns around comparison to baselines, ablation studies, generalization capabilities, and calibration. I believe that after incorporating the proposed changes in the author’s general response, the paper warrants being accepted. The problem formulation is important, the paper proposes new evaluation metrics beyond just success rate which is important as well (and they discuss this in detail in both the paper and rebuttal), and while I share the concerns of Reviewer 9hBH that the proposed solution is heuristic in nature and has quite a few components, empirically the system performs well and is a solid basis for further research to build on. Also, I believe conformal techniques can also be applied to CURE, and I encourage the authors to investigate this for the camera-ready.